# Structural remodelling of the carbon–phosphorus lyase machinery by a dual ABC ATPase

Søren K. Amstrup[1,3], Sui Ching Ong[1], Nicholas Sofos[1,3], Jesper L. Karlsen[1], Ragnhild B. Skjerning[1], Thomas Boesen [2], Jan J. Enghild [1], Bjarne Hove-Jensen[1] & Ditlev E. Brodersen [1] ✉

In *Escherichia coli*, the 14-cistron *phn* operon encoding carbon-phosphorus lyase allows for utilisation of phosphorus from a wide range of stable phosphonate compounds containing a C-P bond. As part of a complex, multi-step pathway, the PhnJ subunit was shown to cleave the C-P bond via a radical mechanism, however, the details of the reaction could not immediately be reconciled with the crystal structure of a 220 kDa PhnGHIJ C-P lyase core complex, leaving a significant gap in our understanding of phosphonate breakdown in bacteria. Here, we show using single-particle cryogenic electron microscopy that PhnJ mediates binding of a double dimer of the ATP-binding cassette proteins, PhnK and PhnL, to the core complex. ATP hydrolysis induces drastic structural remodelling leading to opening of the core complex and reconfiguration of a metal-binding and putative active site located at the interface between the PhnI and PhnJ subunits.

Bacteria have evolved elaborate mechanisms to extract essential macronutrients, such as phosphorus, sulphur, nitrogen, and carbon, from their environments. In the Gram-negative *Escherichia coli*, phosphate limitation induces the Pho regulon and expression of the 14-cistron *phn* operon (*phnCDEFGHIJKLMNOP*), enabling uptake and breakdown of phosphonates as an alternative source of phosphorus[1–3]. Phosphonates are structurally similar to phosphate esters and other primary metabolites but contain a highly stable carbon–phosphorous (C–P) bond, which is resistant to hydrolytic cleavage[4]. Phosphonates can therefore function as inhibitors and transition state analogues and are highly useful in industry as detergents, herbicides (e.g., glyphosate/ RoundUp®) and antibiotics[5]. The *phn* operon encodes an ATP-binding cassette (ABC) importer (PhnCE) and periplasmic binding protein (PhnD) for phosphonate uptake[6], a transcriptional regulator (PhnF)[7], in addition to the complete enzymatic machinery (PhnGHIJKLMNOP) required for degradation and incorporation into general metabolism via 5-phospho-α-D-ribosyl-1-diphosphate (PRPP)[8–11]. The C–P lyase pathway has a remarkably wide substrate spectrum and is capable of

handling aliphatic as well as bulky aromatic phosphonates[10,12,13]. For this reason, and due to the many applications of phosphonates, there is a strong, fundamental interest in understanding the mechanism of phosphonate breakdown by C–P lyase. Detailed functional and mechanistic studies of individual enzymatic subunits encoded by the *phn* operon has led to a proposed reaction mechanism in which the phosphonate moiety is initially coupled to either ATP or GTP through displacement of the nucleobase in a reaction catalysed by PhnI and in the presence of PhnG, PhnH, and PhnL[8]. PhnM was shown to release pyrophosphate from the resulting 5′-phosphoribosyl-α−1-phosphonate to allow for PhnJ to cleave the C–P bond via a strictly anaerobic S-adenosyl methionine (SAM)-dependent glycine-radical reaction mechanism. Finally, the combined action of PhnP and PhnN converts the resulting cyclic ribose into PRPP (Supplementary Fig. 1)[9,10].

Due to the complexity of the reaction, the large number of protein subunits and complexes involved, and the anaerobic requirement, however, it has not yet been possible to obtain a complete, mechanistic understanding of the C–P lyase pathway. The crystal structure of a

[1]Department of Molecular Biology and Genetics, Aarhus University, Universitetsbyen 81, DK-8000 Aarhus C, Denmark. [2]Interdisciplinary Nanoscience Centre (iNANO) Aarhus University, Gustav Wieds Vej 14, DK-8000 Aarhus C, Denmark. [3]Present address: Novo Nordisk Foundation Center for Protein Research, University of Copenhagen, Blegdamsvej 3B, DK-2200 Copenhagen N, Denmark. ✉e-mail: deb@mbg.au.dk

core enzyme complex, Phn(GHIJ)$_2$, revealed a symmetrical hetero-octamer consisting of a compact, central tetramer of the PhnI and PhnG subunits each interacting individually with the more distal PhnJ and PhnH subunits[14]. An intriguing feature of this structure is that the site of the Fe$_4$S$_4$ cluster, presumably required for radical formation and enzyme activation, is located at a significant distance (30 Å) from a glycine residue in PhnJ, which was shown by deuterium exchange studies to be the stable location of a radical between enzymatic turn-over cycles[15]. The crystal structure also revealed another putative active site at the interface of PhnI and PhnJ containing a Zn$^{2+}$ ion coordinated by two conserved His residues in PhnI, both demon-strated to be essential for growth on phosphonate. Finally, it was found that of the two non-transporter ABC modules encoded by the *phn* operon, PhnK and PhnL, a single subunit of PhnK can bind to PhnJ via a conserved helical region known as the central insertion domain (CID)[14]. In bacteria, ABC modules are mostly associated with metabolite importers where they interact as dimers with a transmembrane domain to bind and hydrolyse ATP, thus generating the energy and movement required for transport against a concentration gradient[16]. The general dimeric structure of ABC modules thus offered little towards understanding the function of the single PhnK subunit bound to the C–P lyase core complex. Moreover, this discrepancy suggested that the observed structure might not represent a true functional state of the enzyme and did not provide a rationale for the presence of the other, essential ABC module, PhnL. In this work, we show genetically and enzymatically that the ATPase activity of both PhnK and PhnL are required for *E. coli* growth on phosphonates and use cryogenic elec-tron microscopy (cryo-EM) to reveal a double dimer of the two sub-units on C–P lyase core complex. Structural analysis under ATP turnover conditions further reveal that hydrolysis leads to opening of the core complex exposing and rearranging the Zn$^{2+}$ active site located between the PhnI and PhnJ subunits.

## Results

### A single PhnK subunit binds to the C–P lyase core complex in a disordered state not compatible with ATP hydrolysis

To better understand the interaction of ABC subunits with C–P lyase and the role of these modules in catalysis, we initially purified the PhnK-bound core complex from a plasmid encoding PhnGHIJK with His-tagged PhnK as previously described[14] and determined a high-resolution structure of the purified C–P lyase complex by cryo-EM (Supplementary Fig. 2a and Supplementary Fig. 3). An initial map generated using 901,800 particles resulted in a structure with an overall resolution of 2.2 Å (using the gold standard Fourier shell correlation (FSC) 0.143 cut-off criterion[17]) of the C–P lyase core complex bound to a single PhnK subunit as previously observed (Fig. 1a)[14,16]. PhnK displayed a significantly lower resolution than the core PhnGHIJ complex, so we used 3D classification with signal subtraction to focus the refinement on PhnK and obtained a final class consisting of 50,323 particles at a resolution of 2.6 Å. This map had improved density for PhnK and allowed for tracing of the complete fold of the protein (Fig. 1a, Supplementary Fig. 3, and Supplementary Table 1). Variability analysis revealed several dis-tinct positions of PhnK representing a hinge-like motion of the ABC domain around the site of interaction (Fig. 1b)[18,19]. The specific interaction of PhnK with the CID of PhnJ is driven primarily by electrostatic interactions involving negatively charged residues on PhnJ (Glu149, Asp226, Asp228, and Asp229) and a positive patch on PhnK (Arg78, Arg82, and Arg116) (Fig. 1c, d). These interactions are further stabilised by hydrophobic interaction between PhnJ Tyr158 and PhnK Tyr118. PhnK adopts a classical nucleotide-binding domain (NBD) fold similar to the cytoplasmic domains of ABC transporters and contains all conserved catalytic motifs (Walker A/B, Q loop, ABC signature, and H switch, Fig. 1d, e), several of which are directly involved in ATP binding and hydrolysis[20]. In a subset of

3D classes, additional EM density was visible near the P-loop of PhnK (Fig. 1e and Supplementary Fig. 2b). The density overlaps partially with that of ATP inside NBD modules in ABC transporters (Fig. 1f), but the lack of high-resolution features suggests that any nucleotide binding is disordered and consequently that PhnK is not in a cata-lytically competent state. This is consistent with the general understanding that ABC modules are only active in their dimeric state. In summary, we conclude that when a single PhnK subunit binds to the C–P lyase core complex, it is structurally disordered and cannot engage in ATP hydrolysis. Moreover, and in contrast to previous observations[16], we observe no significant structural chan-ges in the PhnGHIJ core complex, including the putative Fe$_4$S$_4$ cluster site, upon binding of a single subunit PhnK (rmsd = 0.5 Å, Supplementary Fig. 2c).

### PhnK and PhnL associate as a double dimer to the C–P lyase core complex

To capture PhnK in a stable, nucleotide bound conformation, we introduced the E171Q mutation in the Walker B motif (Fig. 1e), which is known to allow binding, but not hydrolysis of ATP in ABC modules[21]. We then purified the C–P lyase complex as expressed from a plasmid encoding the full enzymatic machinery (PhnGHIJKLMNOP) using a C-terminal double Strep-tag on PhnK and in the presence of the non-hydrolysable ATP analogue, β,γ-imidoadenosine 5′-triphosphate (AMPPNP). Structure determina-tion by cryo-EM single-particle analysis at 2.1 Å resolution (59,737 particles, C2 symmetry, Supplementary Fig. 4 and Supplementary Table 1) revealed the core complex bound to a dimer of PhnK in a conformation reminiscent of the ATP-bound state of ABC trans-porters (Fig. 2a, b, red subunits). At the interface of the two PhnK subunits, clear density for AMPPNP allowed for modelling of a nucleotide in both active site pockets (Fig. 2c and Supplementary Fig. 5a). Inside PhnK, the adenosine nucleotide binds in a very similar way to what has been observed for membrane-associated ABC modules (Supplementary Fig. 5d). No structural differences are observed between the C–P lyase core complex when associated with the ATP-bound, dimeric state of PhnK compared to when a single PhnK subunit is bound (rmsd = 0.3 Å, Supplementary Fig. 5e). Remarkably, this structure also revealed two copies of an additional protein situated on top of the PhnK subunits, which we could identify as PhnL by mass spectrometry and inspection of the EM density (Fig. 2a, b, yellow subunits, and Supplementary Fig. 6a). PhnL has an NBD fold like PhnK and but differs by lacking a con-served C-terminal domain also found in the ABC modules of trans-porters and by the presence of a long β hairpin near the N terminus that results from extension of β1 and β2 (Fig. 2a, b, β-extension, green). The association of a PhnL subunit with each of the two PhnK subunits thus generates a double dimer of ABC subunits, both of which are in the same orientation with respect to each other and the core complex and thus potentially capable of nucleotide binding. In this hetero-dodecameric, 327 kDa complex, PhnL is in an open ADP-like state with no visible density for a nucleotide. The interaction between PhnK and PhnL is mediated by a wide range of specific interactions, mostly via the extended C terminal domain of PhnK, which contains two α helices (α8 and α9) that facilitate most con-tacts with PhnL, including its extended β hairpin (Fig. 2d–f). The interactions are primarily ionic with the binding region (top side) of PhnK having an overall negative charge (residues 231–247) and with PhnL (underside) being positively charged (Supplementary Fig. 6b). In addition, there is π–π stacking between PhnK Trp29 and PhnL Arg82 (Fig. 2d). The Arg in PhnL is fully conserved in orthologues, while the Trp in PhnK is functionally conserved for its stacking ability (Supplementary Fig. 6c). In summary, we conclude that both non-transporter ABC subunits encoded by the *phn* operon, PhnK, and PhnL, can bind simultaneously to the C–P lyase core complex in

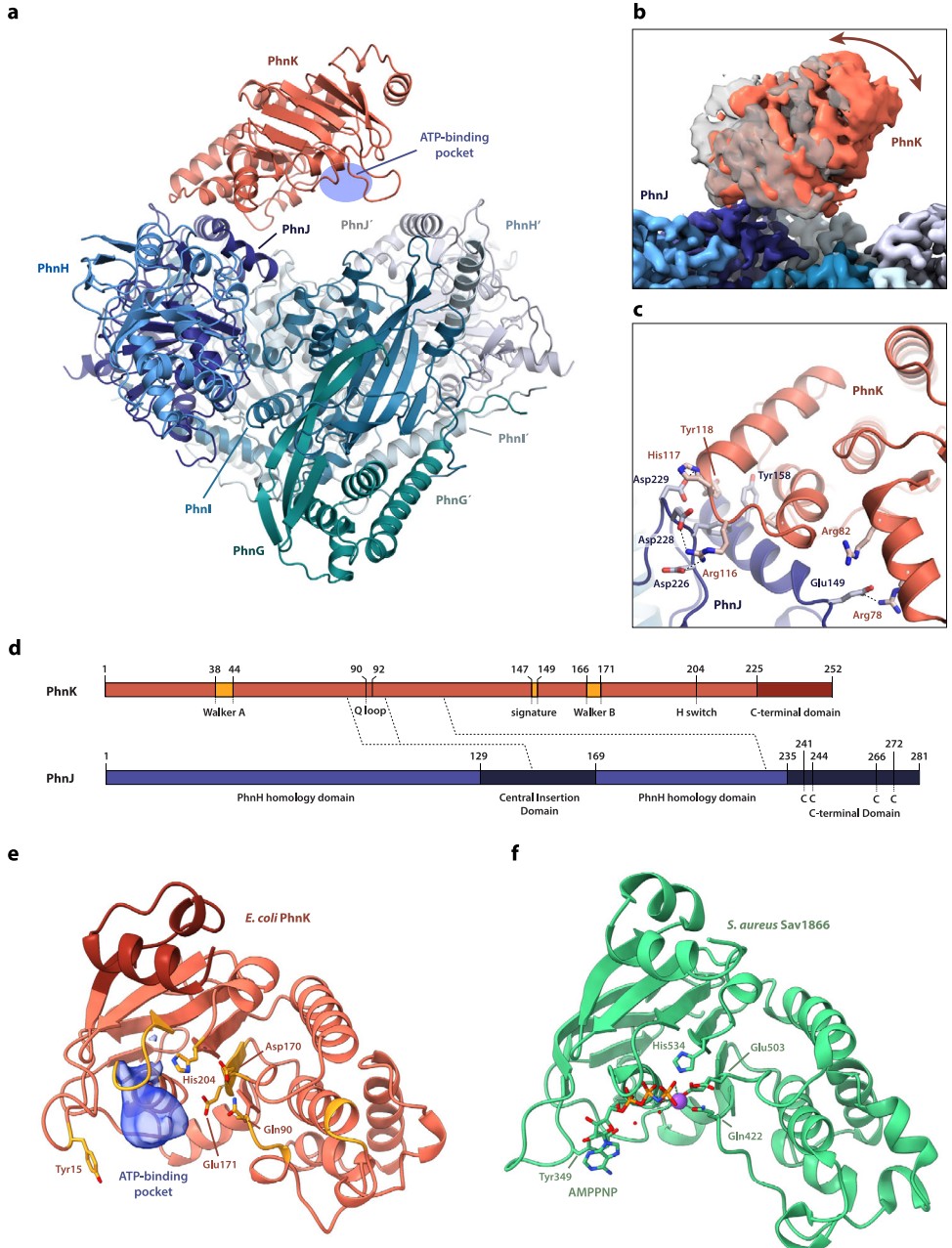

**Fig. 1 | The C-P lyase core complex can bind a single, flexible PhnK subunit.**
**a** Overview of the structure of the C–P lyase core complex (PhnGHIJ, blue/green) bound to a single subunit of PhnK (red) with the names of individual subunits indicated. The ATP-binding pocket of PhnK is shown with a blue ellipse. **b** Surface representation of the EM density showing the extent of the tilting motion observed in PhnK. The two extreme states are shown in grey and red, respectively. **c** Close-up of the interaction between PhnK (red) and the PhnJ central insertion domain (CID, blue) with relevant residues shown as labelled sticks and interactions with dashed lines. **d** Overview of the domain structure of PhnK (top, red) and PhnJ (bottom, blue) with residue numbers indicated. For PhnK, the location of the core ABC motifs (Walker A and B, Q loop, signature motif, and H switch) are indicated in orange and the C-terminal domain in dark red. For PhnJ, the features that distinguish the

protein from PhnH with which it is homologous (the Central Insertion Domain, CID, residues 129–169, and the C-terminal Domain, CTD, residues 235–281) are shown in darker blue. The four Cys residues near the C-terminus involved in Fe$_4$S$_4$ cluster binding are indicated with C. Interactions between the proteins are shown with dashed lines. **e** Close-up view of PhnK with catalytically important residues (Tyr15, A loop; Gln90, Q loop; Asp170/Glu171, Walker A motif; His204, H loop) shown as sticks. An extra density observed in several 3D classes at the nucleotide binding site is shown in blue. **f** Structure of the *S. aureus* Sav1866 multidrug transporter ABC domain (2ONJ)[24] in the same orientation with the nucleotide (AMPPNP) and catalytically important residues (Tyr349, A loop; Gln422, Q loop; Glu503, Walker A motif; His534, H loop) as sticks and the Mg$^{2+}$ ion as a purple sphere.

a double dimer conformation with PhnK in the ATP-bound state and PhnL in an open state compatible with ATP binding upon closure. To our knowledge, this type of interaction between two ABC dimers has not been observed before and therefore represents a hitherto unrecognised functional mode of ABC nucleotide-binding domains.

## Binding and ATP hydrolysis by both PhnK and PhnL are essential for phosphonate utilisation in vivo

To probe the importance of the specific interactions between PhnK and the core complex as well as the functional integrity of PhnK and PhnL for phosphonate breakdown, we used genetic complementation to test if alteration of specific residues would affect the ability of *E. coli*

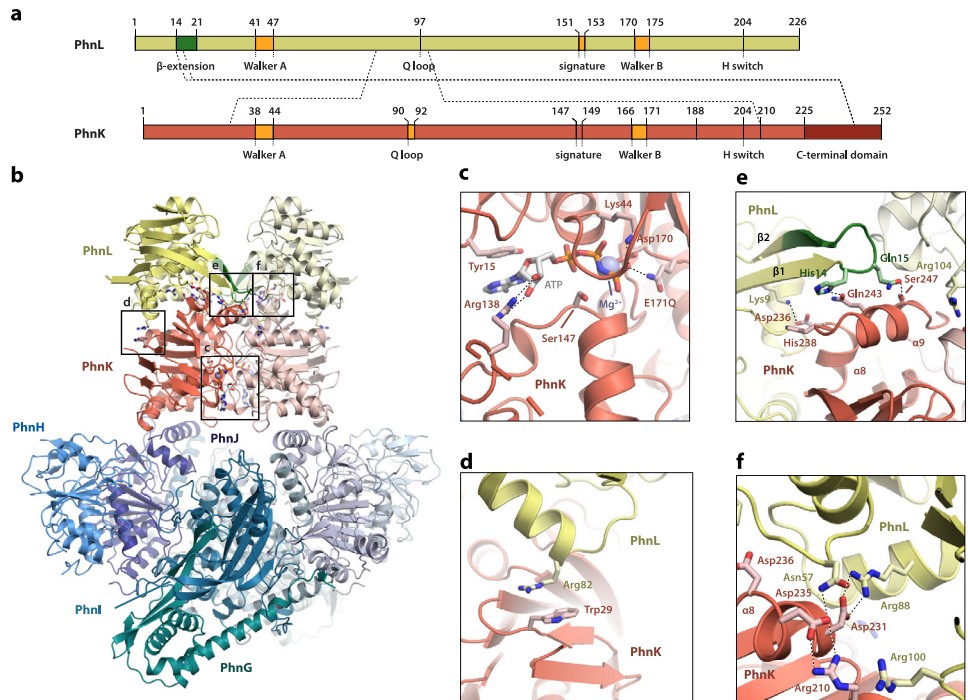

**Fig. 2 | PhnGHIJKL binds a double dimer of PhnK and PhnL. a** Top, an overview of the domain structure and sequences features of PhnL (top, yellow) and PhnK (bottom, red) with residue numbers indicated. The core ATPase motifs (Walker A and B, Q loop, signature motif, and H switch) are indicated in orange, the β hairpin extension of PhnL (residues 7–14) is shown in green and the C-terminal domain of PhnK is shown in dark red. Interactions between the proteins are shown with dashed lines. **b** An overview of the structure of PhnGHIJKL with the C-P lyase core complex PhnGHIJ in shades of blue/green and corresponding light colours for the other half, the PhnK dimer in red/light red, and the PhnL dimer in yellow/light yellow. Boxes indicate the approximate location of the close-up views (**c**–**f**). **c** Details of the PhnK ATP binding site with the AMPPNP (white/orange) shown alongside relevant, interacting side chains (labelled sticks), and the $Mg^{2+}$ ion as a blue sphere. **d** Stacking interaction observed between PhnK Trp29 and PhnL Arg82. **e** Interaction between the PhnK C-terminus (helices α8 and α9, residues 236–247) and the PhnL β hairpin extension (residues 7–14). **f** Charged interactions observed between the PhnK C-terminal region (residues 210–236) and PhnL.

to grow on phosphonate as a sole source of phosphorus (Fig. 3a). *E. coli* strain HO1488 (Δ*phnHIJKLMNOP*), which lacks the ability to grown on phosphonate, can be rescued by introduction of a plasmid-borne copy of the *phn* operon (pSKA03, see Methods for details) and thus be used to test the functional effects of mutants of the individual proteins. Initially, we could show that the inability of the HO1488 strain to grow on minimal media containing either methylphosphonate or 2-aminoethylphosphonate could be genetically complemented by introduction of pSKA03 (Fig. 3a). Disruption of key residues located at the PhnK-PhnJ interaction interface (PhnJ E149A, Y158A or PhnK R78A/ R82A) or inhibition of ATPase activity of either PhnK or PhnL by mutation of the catalytically active glutamate (PhnK E171Q or PhnL E175Q)[21] again abolished growth on phosphonate as a sole source of phosphorus. Together, this demonstrates that both interaction of PhnK and PhnL with the core complex as well as the ATPase activity of these subunits are required for phosphonate breakdown in vivo.

Since ATP hydrolysis is essential for the ability of *E. coli* to grow on phosphonate, we next asked if the C–P lyase core complex bound to PhnK and PhnL, Phn(GHIJKL)₂, is active in ATP hydrolysis in vitro. To answer this, purified wild type, as well as PhnK and PhnL inactivated complexes (Supplementary Fig. 7a, b), were individually analysed for ATPase activity by incubation with ATP and $Mg^{2+}$ followed by chromatographic separation of nucleotides (Fig. 3b) as well as by a coupled enzymatic ATPase assay (Fig. 3c). Both techniques revealed a clear ATPase activity for the wildtype PhnGHIJKL complex, while the Phn(GHIJ)₂K complex showed negligible activity (Supplementary Fig. 7c), consistent with a requirement for the presence of a complete ABC dimer for ATPase activity. For both of the single PhnK E171Q and PhnL E171Q mutants we observed a significantly reduced ATPase activity, which was slightly more pronounced when PhnK was inactive, while for the double PhnK E171Q/PhnL E171Q mutant, activity was

completely absent (Fig. 3c). In summary, our functional data clearly show that the ATPase activity of C–P lyase requires a dimer of ABC subunits and that both PhnK and PhnL are required for maximum activity both in vivo as well as in vitro. Moreover, both subunits have to be able to bind to the C–P lyase core complex in order for phosphonate breakdown to take place in vivo.

## ATP turnover by PhnK induces large-scale conformational changes in the C–P lyase core complex

To capture potential, additional conformational states of the complex resulting from the ATPase activity in PhnK and PhnL, we next purified wild type Phn(GHIJKL)₂ complex in the presence of ATP and EDTA (to initially prevent ATP hydrolysis by PhnK), then added excess $Mg^{2+}$ to initiate the reaction immediately before spotting onto cryo-EM grids and plunge freezing. Cryo-EM data collection and processing under these conditions yielded structures of several functional states including a high-resolution structure of the Phn(GHIJKL)₂ complex extending to 1.9 Å using a final set of 222,056 particles and by imposing C2 symmetry (Supplementary Figs. 8 and 9 and Supplementary Table 1). This state is essentially identical to the structure determined in the presence of AMPPNP, however, close inspection of the PhnK active site in the high-resolution EM map revealed that it contains ADP and $P_i$ and consequently represents a true ATP post-hydrolysis state (Supplementary Fig. 5b). We also observe binding of ATP to both nucleotide binding sites in PhnL, but despite this, the dimer is still in an open, ADP-like state (Supplementary Fig. 10a).

3D classification of the data collected under ATP turnover conditions yielded structures of two additional states of the core complex bound to PhnK and in some cases PhnL (Fig. 4a, b and Supplementary Figs. 8 and 9, and Supplementary Table 1). In both structures, the density for PhnK is significantly poorer than in the stable PhnL-bound

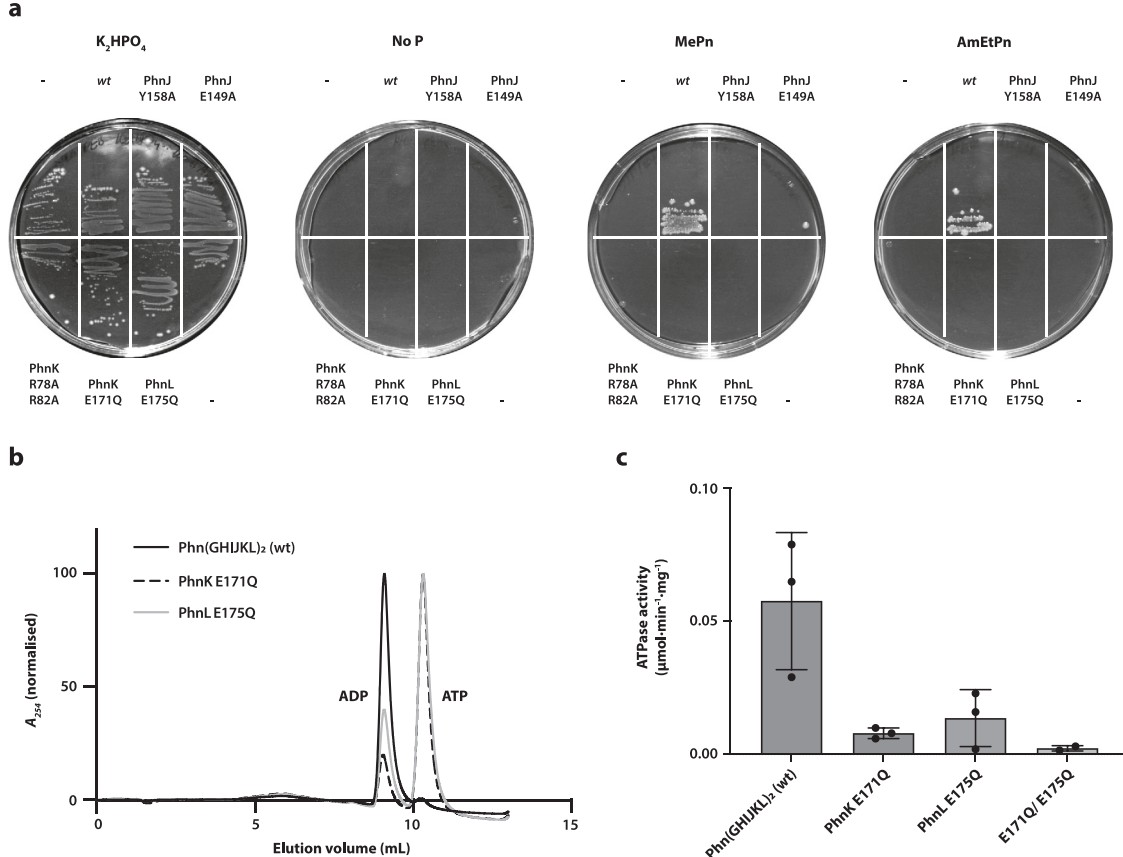

**Fig. 3 | ATP hydrolysis by PhnK and PhnL is required for phosphonate utilisation in vivo. a** In vivo functional assay using the *E. coli* HO1488 strain (Δ*phnHIJKLMNOP*) grown either without plasmid (−) or complemented with pSKA03 containing the entire *phn* operon including the *pho* box and including the following specific mutations: wt (none), PhnJ Y158A, PhnJ E149A, PhnK R78A/R82A, PhnK E171Q, and PhnL E175Q. Cells were plated on MOPS minimal agar plates[44] containing either K₂HPO₄, 2-AmEtPn, MePn, or no added phosphorus source (no P) as indicated. The plates are representative of two repetitions. Source data are provided as a Source Data file. **b** ATPase activity using purified Phn(GHIJKL)₂

wildtype, Phn(GHIJKL)₂-PhnK E171Q (PhnK E171Q), and Phn(GHIJKL)₂-PhnL E175Q (PhnL E175Q) measured by separation of nucleotide species by ion exchange chromatography after overnight incubation with ATP. **c** Coupled assay ATPase activity (μmol/min/mg) using Phn(GHIJKL)₂ wildtype (wt), Phn(GHIJKL)₂-PhnK E171Q (PhnK E171Q), Phn(GHIJKL)₂-PhnL E175Q (PhnL E175Q), and the double mutant Phn(GHIJKL)₂-PhnK E171Q-PhnL E175Q (E171Q/E175Q). Bars show the mean from three independent reactions and error bars show standard deviation from the mean. One measurement with negative value was excluded from E171Q/E175Q. Source data are provided as a Source Data file.

state, suggesting it is more flexible and possibly therefore that PhnL acts to stabilise the closed conformation of PhnK (Supplementary Fig. 10b, c). In one of these conformations (closed, 2.0 Å, 81,605 particles, C2 symmetry), the PhnK dimer is in the closed ATP-bound state and resembles that found in the Phn(GHIJKL)₂ structures while the density for PhnL is absent. Inspection of the active site reveals clear density for both ATP and Mg²⁺ suggesting that this is a pre-hydrolytic ATP-bound state (Fig. 4a and Supplementary Figs. 5c and 10d, e). In the other structure (open, 2.6 Å, 31.280 particles, C1 symmetry), one of the two PhnK subunits has moved away from the closed position by 25 Å while the other has stayed in place (Fig. 4b).

Due to the tethering of PhnK to the core complex, this movement also results in pulling one set of PhnJ and PhnH subunits away from the rest of the core and in that process exposing the Zn²⁺-binding and putative active site located at the PhnI-PhnJ interface (Fig. 4b). Both PhnK subunits appear flexible (similar to the structure with a single PhnK subunit bound) and display a significantly lower local resolution than the rest of the complex. Due to this flexibility, PhnL could not be modelled in the open state, but inspection of the 2D classes confirms that it is present and possibly in a closed, ATP-like conformation (Supplementary Fig. 10d, e). This suggests that the opening of the PhnK dimer might be the trigger that activates PhnL ATP hydrolysis. Due to flexibility, is not clear whether PhnL is also present in the closed (post ATP hydrolysis) state.

Given that the core complex has appeared monolithic in all previous structures, the concomitant movement of PhnK, PhnJ, and PhnH in the open state represents an architectural rearrangement in C–P lyase with significant functional implications. In the closed state, both PhnI subunits and one PhnJ subunit interact tightly to form a Zn²⁺-binding site located at the interface of the three proteins (the His site)[14]. In addition to two histidine residues originating from PhnI (His328, His333), the Zn²⁺ ion is coordinated by three water molecules and an oxygen from an unknown ligand, which was also observed previously by X-ray crystallography, in an octahedral arrangement (Fig. 4c, closed, and Supplementary Fig. 11a)[14]. Based on inspection of the EM density we were able to model the ligand as 5-phospho-α-D-ribose-1,2-cyclic-phosphate (PRcP, Supplementary Figs. 1 and 11b)[8,10]. While this is an intermediate of the C–P lyase pathway, protein expression was carried out in the absence of phosphonates and we, therefore, believe it could have been formed in a side reaction during overexpression in the absence of phosphonate and was carried along during purification. Due to the minute amounts of the compound present (two molecules per 350 kDa complex), attempts to identify it by UPLC-QTOF-MS differential analysis were not successful. The ligand is tightly bound at this site with specific interactions between the ribose moiety and PhnJ Arg107 and Gln124, contacts between the 5' phosphate and a loop (residues 47–50) in PhnJ and finally between two oxygen atoms of the 1,2-cyclic phosphate, PhnJ His108, and Zn²⁺

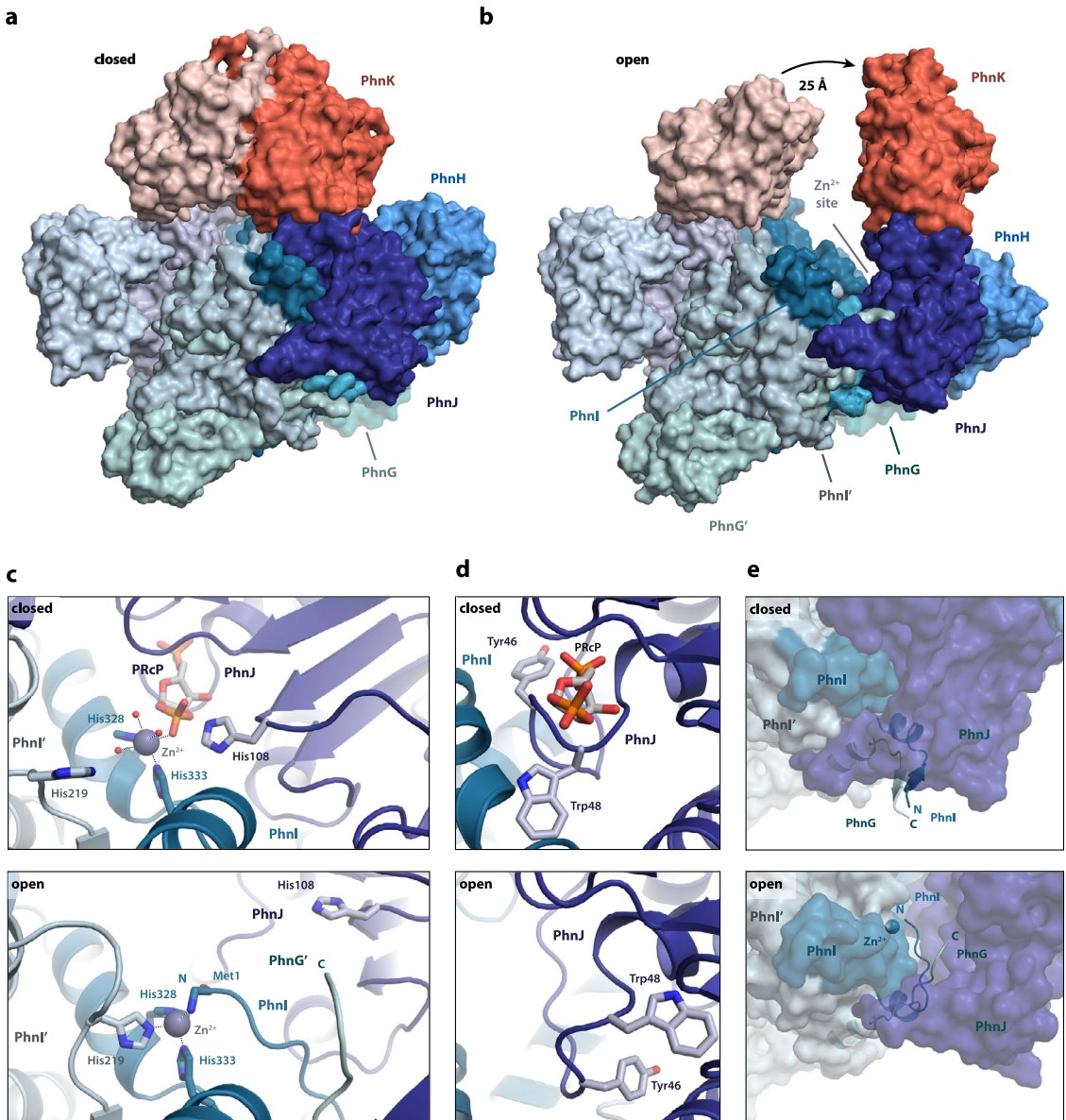

**Fig. 4 | ATP hydrolysis by PhnK leads to opening of the C–P lyase core complex.** Overview of the structural changes taking place in the Phn(GHIJK)$_2$ complex between the closed (**a**) and open (**b**) states. The Phn(GHIJK)$_2$ complex is shown as a surface representation with the C–P lyase core complex in shades of blue/green and PhnK in red/light red. The arrow indicates the extent of opening at the Zn$^{2+}$ site. **c** Details of the presumed active site at the interface between PhnI and PhnJ in the closed (top) and open (bottom) states. The Zn$^{2+}$ ion is shown as a grey sphere with coordination geometry indicated and relevant, interacting residues with labelled sticks. **d** Details of the active site pocket with a bound molecule of 5-phospho-α-D-ribose-1,2-cyclic-phosphate (PRcP) as modelled in the closed (top) and open (bottom) states. **e** Coordinated movement of the PhnI N terminus and PhnG C terminus from the surface of the PhnGHIJK complex in the closed state (top) to the active site in the open state (bottom).

(Supplementary Fig. 11b). We also note that there is a large cavity next to what would correspond to the position of the phosphonate atom of a natural substrate in this orientation, which supports that the bound compound mimics a reaction intermediate and could explain the wide substrate specificity of C–P lyase (Supplementary Fig. 11c).

On the contrary, no ligand density is observed in the Zn$^{2+}$ site in the open state, which has undergone a rearrangement, whereby the loop of PhnJ that interacts with the 5′ phosphate of the ligand in the closed state has been completely remodelled (Fig. 4d), allowing His219 from a nearby PhnI chain and the N-terminal NH$_2$ group of PhnI to complete the Zn$^{2+}$ coordination sphere, which is now tetrahedral (Fig. 4c, open, and Supplementary Fig. 11d). Meanwhile, His108 of PhnJ, which interacts with the 1,2-cyclic phosphate group of PRcP in the closed state has moved ~11 Å away from the metal ion. There are very few internal rearrangements in PhnJ and PhnH, which appear to move

as near rigid bodies (rmsd = 0.6 Å between open and closed state structures of these subunits) except the PhnJ loop 37–49 that lines the active site pocket and appears to help accommodate the rearrangements around the Zn$^{2+}$ ion. This also means that there is no change in the structural relationship and distance between the putative Fe$_4$S$_4$ cluster site involving PhnJ cysteine residues 241, 244, 266, and 272 and the proposed radical site at PhnJ Gly32. In the open state of the core complex, Gly32 is buried inside PhnJ about 10 Å from the surface near the active site cavity and 31 Å from the Zn$^{2+}$ ion (Supplementary Fig. 11e).

The extended C terminus of PhnG, which in the closed state is located on the outside of the C–P lyase core complex and forms a tight interactions with the N terminus of PhnI using a small antiparallel β motif, is carried along with the PhnI N terminus as it relocates to the Zn$^{2+}$ site and gets tucked away in a pocket in PhnJ (Fig. 4e). Together,

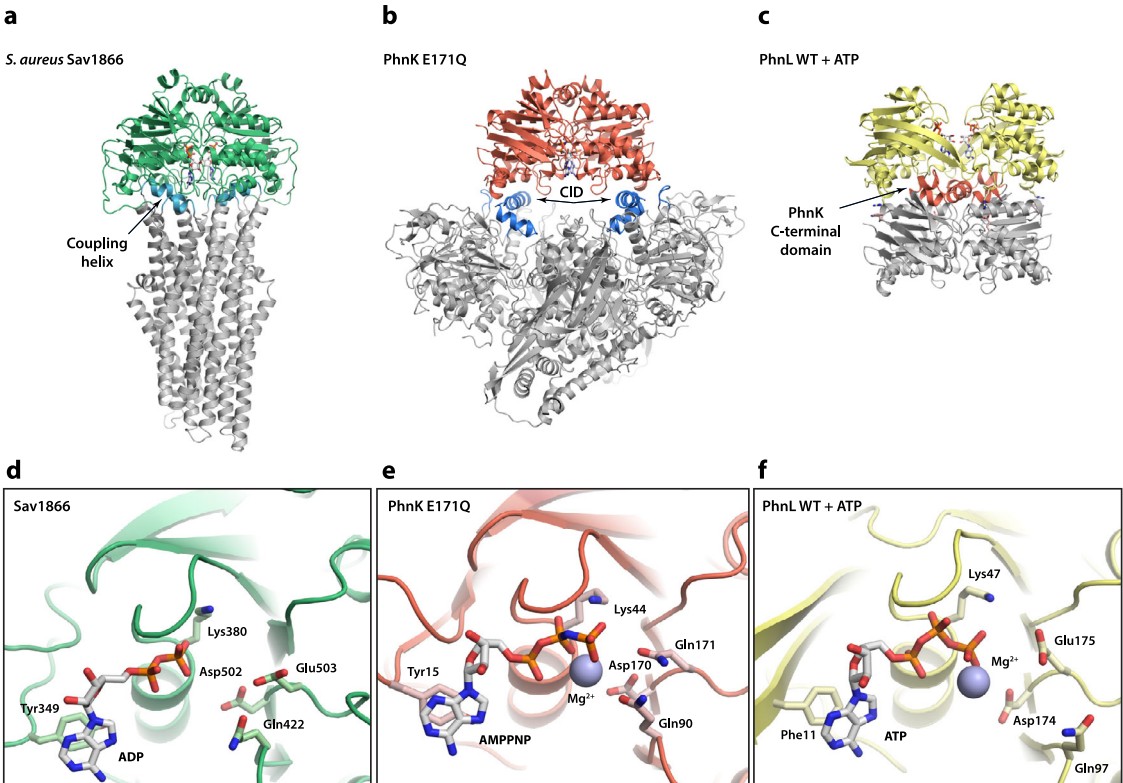

**Fig. 5 | Comparison to the ABC transporters. a** Overview of the structure of the *S. aureus* Sav1866 ABC transporter in the ADP-bound conformation with the ABC module in green and transmembrane domain in grey, except for the coupling helix, responsible for the interaction between the two, which is shown in cyan (2HYD)[24] **b** Overview of the structure of the C-P lyase core complex with a dimer of PhnK E171Q in the ATP bound conformation with the PhnK dimer in red and the C-P lyase core complex in grey except for the PhnJ Central Insertion Domain (CID), responsible for PhnK binding, which is shown in blue. **c** Overview of the structure of the PhnK ABC dimer bound to PhnL in the open conformation with the PhnL dimer in yellow and PhnK in grey except for the C-terminal region, responsible for the interaction, which is shown in red. **d** Details of the *S. aureus* Sav1866 ABC transporter ATP binding site (bound to ADP). **e** Details of the PhnK ATP binding site (bound to AMPPNP). **f** Details of ATP bound to the PhnL active site.

the PhnI and PhnG termini, therefore, undergo concerted movements, in both cases of more than 30 Å, a structural change that likely has several important, functional implications. We note that the involvement of both chains of PhnI in $Zn^{2+}$ coordination in the open state also explains, along with the symmetry requirement for binding an ABC dimer, why the C–P lyase core complex must be a dimer of PhnGHIJ, thus solving a long-standing conundrum concerning the reason for the higher-order structure of C–P lyase.

## Discussion

In this work, we present structures of several states of the *E. coli* C–P lyase core complex bound to the ABC modules PhnK and PhnL showing that neither their interaction with the core nor ATP binding per se alter the conformation of the enzyme. This is different from the classical role of ABC modules in transporters, where ATP binding is known to induce a more compact state as well as changes in the transmembrane segment resulting in eversion and the outward open state[22]. It thus appears that the stable ground state of the C–P lyase core complex (i.e., that observed in the absence of the ABC modules) is compatible with the ATP-bound conformation of PhnK. We also show that the presence of a dimer of PhnK on the C–P lyase core complex results in association of a similar dimer of the homologous ABC module, PhnL, in a double ATPase arrangement. And importantly, we could show that both in vivo and in vitro, the activity of both ABC modules is required for ATP turnover. While it is still not clear why and when ATPase activity is required during phosphonate breakdown, this observation raises several questions regarding activation of the ATPases and whether ATP hydrolysis happens simultaneously or sequentially in the two sets of proteins. Structure determination under

ATP turnover conditions allowed us to capture a closed, post-hydrolysis structure for PhnK as bound to the core complex and $ADP + P_i$. For the ABC transporters, this conformation is difficult to capture due to fast release of $P_i$ and energy upon ATP hydrolysis[23], so this result might suggest that PhnL serves to control phosphate release from PhnK and therefore potentially also provides an explanation for the requirement for two types of ABC modules.

Both PhnK and PhnL bind in the same orientation relative to their partners, the C–P lyase core complex and PhnK, respectively, as the corresponding NBDs bind the transmembrane domains in the ABC transporters, but the structure of the interacting region differs in both cases from the typical coupling helix found in ABC transporters (Fig. 5). In both cases, interaction between the NBD and its partner uses another binding geometry and appears to require special elements, such as the extended β hairpin in PhnL and the C terminal domain of PhnK (Figs. 2a, b and Fig. 5). Interestingly, this C-terminal domain is often found in the ABC modules of transporters, such as the *Staphylococcus aureus* multidrug transporter, Sav1866 (Figs. 5a and 5d)[24]. The double dimer of PhnK and PhnL observed in C–P lyase has to our knowledge not been observed before among ABC modules and therefore significantly expands the repertoire of how these proteins can interact with partners. Moreover, since the C–P lyase core complex is unrelated to the transmembrane segments of transporters, it also indicates that the overall fold of the ABC binding partner is of less importance for interaction than specific interactions at the interface. As hypothesised above, PhnL could potentially function to inhibit $P_i$ release from PhnK post hydrolysis and thus control when the energy generated from ATP hydrolysis is released, likely to break open the core complex as shown. The EM density suggests that the C-terminal

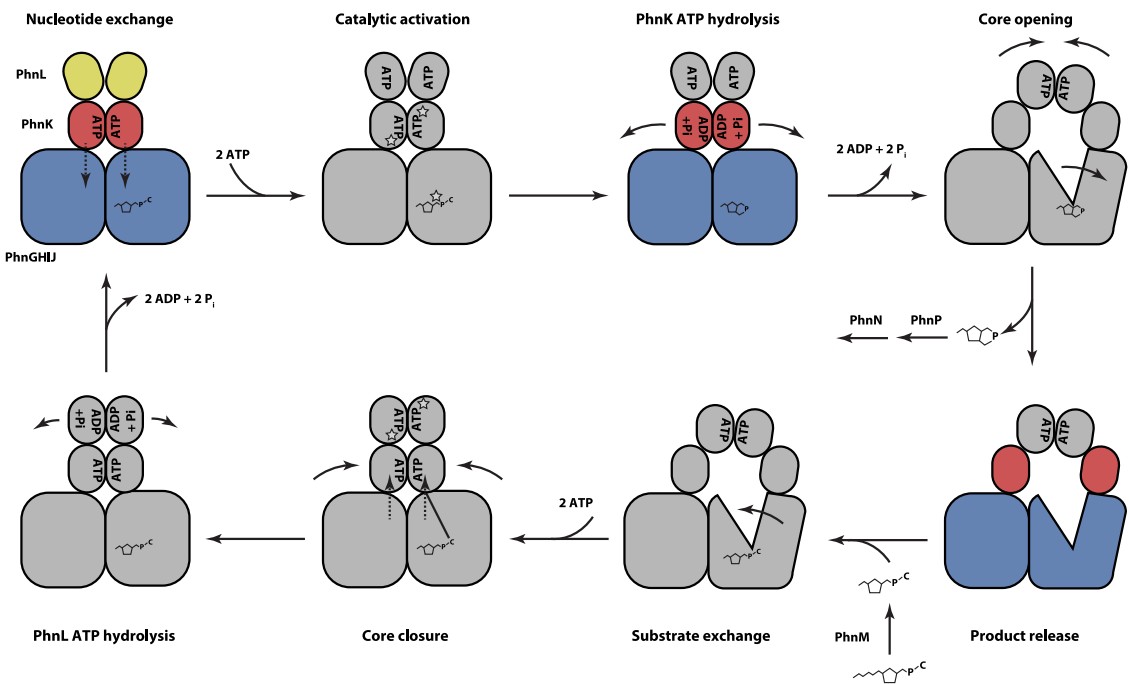

**Fig. 6 | Model for the role of a double ABC module in catalysis by C−P lyase.** A schematic model depicting a possible functional cycle for C−P lyase with known (observed) states in colours and putative states in grey. At the top left, the C−P lyase core complex (PhnGHIJ, blue) binds two dimers of PhnK (red, in the ATP-bound state) and PhnL (partly open) as well as substrate. This complex can bind ATP in PhnL before possibly triggering a reaction in the core complex and ATP hydrolysis and $P_i$ release in the PhnK dimer. PhnK ATP hydrolysis causes opening of the core to allow product release while the PhnL subunits are allowed to form a closed dimer and substrate is exchanged. Following renewed substrate binding, the core closes while bringing the two PhnK subunits in close proximity in an ATP bound state. Finally, PhnL hydrolyses ATP, releases ADP + $P_i$, and separates leading back to the starting position.

region of PhnK becomes more ordered in the presence of PhnL (Supplementary Fig. 10b, c). This effect extends towards helix 6 and the D loop, which could act as a gate to control $P_i$ release. In this model, stabilisation of the PhnK D loop by PhnL would prevent $P_i$ release.

Given that the C−P lyase core complex has appeared monolithic in all previous structures, we believe that the structural rearrangement from a closed to an open conformation is significant and will provide new ways of understanding the complex catalytic process of phosphonate breakdown by this pathway. Presumably, the inaccessible nature of the metal binding site between PhnI and PhnJ prevents ligand dissociation in the closed state. Conversely, this also suggests that opening of the complex and full occupation of the $Zn^{2+}$ ion could induce substrate exchange. And by extension, substrate binding might be a requirement for closure of the core complex, which would explain why there is always density for a ligand near the $Zn^{2+}$ in the closed state[14]. Together, the results presented here thus allow us to propose a model for the structural rearrangements taking place in C−P lyase during a part of the reaction cycle (Fig. 6). Initially, the core complex would be in a closed state with substrate bound, PhnK in the ATP state and PhnL in a semi-open state (Fig. 6, top left). This could trigger a reaction at the active site of the enzyme as well as ATP hydrolysis on PhnK. The two PhnK subunits would then be pushed apart, open the core complex, and allow for closure of PhnL. In this state, the active site is open and substrate/product exchange could take place (Fig. 6, bottom right). Binding of another substrate to the site could then trigger renewed closure of the core and ATP hydrolysis in PhnL, followed by opening of the PhnL dimer.

At this stage, however, it is not clear exactly during what part of the chemical transformation pathway these rearrangements take place, but some evidence suggests it might be the very first step, which has been shown to involve both PhnG, PhnH, PhnI, and PhnL as well as ATP (Supplementary Fig. 1). The reaction mechanism of the bacterial C−P lyase machinery has remained an unresolved conundrum in

biology since its discovery more than 35 years ago[2,13]. The chemical complexity of the pathway, the multitude of interacting protein domains and enzymes, their internal dependencies, and the requirement for anaerobic reaction conditions has proved a challenging combination. However, steady progress has been made in recent years, both from the biochemical and structural side, and we are now slowly beginning to reveal some of the inner secrets of this fascinating molecular system[8,14,15]. To move on from here, we believe it will be important to study the enzyme under anaerobic conditions using gentle expression conditions that allow for natural incorporation of the $Fe_4S_4$ cluster. Moreover, it is possible that the function of C-P lyase is connected to its intracellular localisation. Therefore, to obtain the full picture, we need to get as close as possible to the natural state and study the reaction as it takes place inside cells.

## Methods

### Expression and purification of Phn(GHIJ)₂K

Plasmid pHO575 encoding PhnGHIJK with a C-terminal His-tag on PhnK was introduced into *E. coli* strain HO2735 (Δ*(lac)X74* Δ*phnCDEF-GHIJKLMNOP* 33−30/F lacI^q zzf::Tn10)[14]. The cells were grown in Luria Bertani (LB) media containing 100 µg/mL ampicillin in a shaking incubator at 37 °C until an $OD_{600}$ of 0.5 was reached at which time gene expression was induced using 0.5 mM Isopropyl β-D-1-thiogalactopyranoside (IPTG). Expression was carried out overnight at 20 °C, after which the cells were collected by centrifugation, resuspended in 50 mM HEPES/NaOH pH 7.5, 150 mM NaCl, 20 mM imidazole, 5 mM $MgCl_2$, 3 mM β-mercaptoethanol (BME), 20% (v/v) glycerol, and 1 mM phenylmethylsulfonyl fluoride (PMSF), lysed by sonication and centrifuged at $23,400 \times g$ for 45 min at 4 °C. The supernatant was loaded onto a 5 mL pre-equilibrated His-Trap HP column (Cytiva), washed with 50 mM HEPES/NaOH pH 7.5, 650 mM NaCl, 20 mM imidazole, 5 mM $MgCl_2$, 3 mM BME, and 20% (v/v) glycerol followed by 50 mM HEPES/NaOH pH 7.5, 150 mM NaCl, 20 mM

imidazole, 5 mM MgCl$_2$, 3 mM BME, 20% (v/v) glycerol, and 50 mM imidazole before elution in the same buffer with 300 mM imidazole. Purified samples were loaded onto a 1 mL Source 15Q (Cytiva) pre-equilibrated with 50 mM HEPES/NaOH pH 7.5, 100 mM NaCl, 5 mM MgCl$_2$, and 5 mM BME, washed and eluted using a linear gradient from 100 to 800 mM NaCl over 20 column volumes (CV). Samples containing PhnGHIJK were diluted to ~100 mM NaCl before loading onto a MonoQ column, pre-equilibrated in 50 mM HEPES/NaOH pH 7.5, 100 mM NaCl, 5 mM MgCl$_2$, and 5 mM BME, before elution using a gradient from 100 to 500 mM NaCl over 36 CV. Individual peaks were pooled and concentrated using Vivaspin 20 Ultrafiltration filters before loaded on a Superdex 200 Increase 10/300 GL size exclusion column pre-equilibrated in 50 mM HEPES pH 7.5, 125 mM NaCl, and 5 mM BME. Finally, peak fractions were stored on ice until used for cryo-EM grids.

### Cryo-EM data collection and analysis of Phn(GHIJ)$_2$K

Cryo-EM grids (C-Flat 1.2/1.3 400 mesh copper grids, Protochips) were glow-discharged for 90 s at 10 mA using a GloQube system (Quorum Technologies) immediately before applying 3 µL of purified Phn(GHIJ)$_2$K (1.7 mg/mL), blotting (6–7 s) and plunge-freezing in liquid ethane using a Leica plunge freezer EM GP2 set to 5 °C, 100% humidity, and an ethane temperature of −184 °C. EM data were collected at the Electron Bio-Imaging Centre (eBIC) at Diamond Light Source, UK on a Titan Krios 300 keV with a direct electron detector (Gatan K3), an energy filter operated with a slit width of 20 eV using SerialEM[25] for data acquisition. 17,088 movies were collected with an exposure of 0.2 s per frame over 40 frames with a dose of 1.3 e⁻/Å²/frame corresponding to a total exposure of 52 e⁻/Å², the defocus ranging from −0.7 to −2.0 µm the nominal magnification was set to 135.000 giving a physical pixel size of 0.83 Å in counting mode. Data collection quality was monitored by running RELION-3[26] in streaming mode through relion_it.py.

Frame-based motion correction was performed using RELIONs implementation of MotionCor2[27] while CTF estimation was performed using Gctf[28] within RELION-3. Subsets of the micrographs were used for Laplacian-of-Gaussian picking in RELION-3 before extraction with 8.75 times binning and 2D classification. Good 2D classes were used as templates for template-based picking after which these particles were extracted using 6 times binning and 2D classification was carried out to remove bad particles. Particles were then re-extracted using 4 times binning, followed by initial model generation and 3D classification. Resulting 3D classes resembling PhnGHIJK comprised a stack of 1,329,400 particles. A second round of 3D classification in RELION-3 and heterogenous refinement in cryoSPARC (Structura Biotechnology Inc.)[29] led to a final stack of 901,800 particles. These particles were then moved back into RELION-3.1 where they were further refined using Bayesian polishing, per-particle CTF refinement, estimation of higher-order aberrations, beam tilt and anisotropic magnification[30]. After auto-refinement in RELION-3.1, an FSC resolution with masking of 2.18 Å was reached. The resulting map showed great detail of the core complex (PhnGHIJ), which agreed well with the structure determined previously by X-ray crystallography[14], while the PhnK subunit displayed significantly lower local resolution. To improve the structure of PhnK, 3D classification with signal subtraction and 3D variability was used[18,19] as detailed in Supplementary Fig. 3. Particles from resulting classes showing interpretable density for PhnK were reextracted and used for new 3D auto-refinement runs. The resulting map had a FSC resolution of 2.57 Å and improved local resolution for PhnK. An initial model was constructed by using the published structure of Phn(GHIJ)$_2$ (4XB6)[14] and a predicted model of PhnK from Phyre2[31] for rigid-body modelling using UCSF Chimera[32]. This model was further fitted into the map using Namdinator[33] and manually built in Coot[34] for the regions of the map with higher

resolution whereas ISOLDE[35] was used for regions of lower resolution and interpretability. The final model was refined and validated using Phenix real-space refinement[36,37].

### Cloning of Phn(GHIJKL)$_2$ and variants

pRBS01 for expression of Phn(GHIJKL)$_2$ was constructed using Gibson assembly[38] by amplification of *phnGHIJKLMNOP* from pBW120[39] and insertion of a sequence encoding a TEV-2xStrep-tag at the 3′ end of *phnK* (see Supplementary Tables 2 and 3 for plasmids and primers used in this work). Linearised pET28a was prepared by PCR using primers RBS01 and RBS02, while the TEV-2x-Strep site was prepared from a gBlock sequence using primers RBS03 and RBS04. The two fragments containing the *phn* genes were prepared from pBW120 using primers RBS05 + RBS06 (*phnGHIJK*) and RBS07 + RBS08 (*phnLMNOP*), respectively. Linearised plasmid was mixed with 2–3-fold excess of the three insert fragments followed by addition of Gibson Assembly® Master Mix (New England Biolabs) and incubation at 50 °C for 60 min. Assembled plasmids were transformed into *E. coli* strain NEB 10-β competent cells (New England Biolabs) and plated on LB agar containing 50 µg/mL kanamycin. PhnK/PhnL mutants were constructed by PCR using pRBS01 as template and primer sets SKA01 + SKA02 (PhnK E171Q) and SCO01 + SCO02 (PhnL E175Q). The double mutant was constructed using these two primer sets in sequential PCR mutagenesis reactions.

### Purification of Phn(GHIJKL)$_2$-PhnK E171Q

For cryo-EM analysis, pRBS01-PhnK E171Q was transformed into competent *E. coli* Lemo21 cells (New England Biolabs). Cells were grown in LB media containing 300 µM L-rhamnose, 34 µg/mL chloramphenicol, and 50 µg/mL kanamycin to an OD$_{600}$ of ~0.5 before addition of 1 mM IPTG. The cells were incubated at 37 °C for 3–4 h before harvesting. Collected cells were resuspended in lysis buffer containing 25 mM HEPES/KOH pH 7.5, 125 mM KCl, 5 mM MgCl$_2$, 10% glycerol, 5 mM BME, 1 µg/mL DNase I, 1 mM PMSF, and 5 µM adenosine-5′-[(β-γ)-imido]triphosphate (AMPPNP) and lysed by sonication. The lysate was cleared by centrifugation at 23,400 × *g* for 45 min at 4 °C and loaded onto a 1 mL StrepTrap HP column (Cytiva) which was kept at 8 °C and had been pre-equilibrated in lysis buffer. The column was washed with 10 CV of a StrepTrap wash buffer (25 mM HEPES/KOH pH 7.5, 125 mM KCl, 5 mM MgCl$_2$, 5 mM BME, and 5 mM AMPPNP) before elution with 5 CV of this buffer including 2.5 mM D-desthiobiotin. Relevant fractions were analysed by SDS-PAGE and stored on ice.

### Cryo-EM sample preparation and analysis of Phn(GHIJKL)$_2$-PhnK E171Q

UltrAuFoil 0.6/1 grids were glow discharged (10 mA, 90 s, GloQube, Quorum) immediately before dispensing 3 µL freshly purified protein diluted in StrepTrap wash buffer to a concentration of 2.5 mg/mL followed by blotting (7–9 s) and immediate plunge-freezing in liquid ethane using a Leica plunge freezer EM GP2, adjusted to 5 °C, 100% humidity and an ethane temperature of −184 °C. Cryo-EM data was acquired at the Danish National Cryo-EM Facility (EMBION) – AU node (iNANO, Aarhus University) on a Titan Krios G3i (Thermo Fisher Scientific) operated at 300 keV using a K3 direct electron detector (Gatan), a Bioquantum energy filter (Gatan) with a slit width of 20 eV and EPU (Thermo Fisher Scientific) for automated data collection. Data were collected in counted super resolution mode at a nominal magnification of 130,000 with gain correction on the fly followed by 2× binning in EPU resulting in movies output at physical pixel size (0.647 Å/pixel). Eucentric height was determined for each grid square, and data acquired using a defocus range of −0.5 to −1.4 µm and an electron fluence of 62 e⁻/Å² over 56 frames. Data quality was monitored and initially processed through cryoSPARC Live.

All data were processed inside cryoSPARC. Initially, micrographs were patch motion corrected, followed by patch CTF-estimation and

filtering using various data quality indicators. Particles were initially picked using a gaussian blob picker, after which 4× binned particles were extracted and subjected to 2D classification. Particles from good 2D classes were kept and a manual picking job using a subset of 14 micrographs was initiated where more particles were picked. The collected set of picked particles were used to train the Deep picker, after which the model was used to pick particles the remaining 4502 micrographs.

The data processing strategy is shown in Supplementary Fig. 4. Briefly, a total of 1,530,855 particles were extracted using 4× binning and subjected to 2D classification, following selection of good 2D classes and 3D ab initio reconstruction using five classes and heterogenous refinement using the same classes. A class showing the Phn(GHIJKL)$_2$ structure was subjected to a homogenous refinement and the aligned particles was used for 3D variability analysis[19]. For 3D variability analysis, three orthogonal principal modes were determined with resolution filtered to 5.5 Å, particles from different conformations were separated by clustering the particles using their principal modes into four different clusters. One cluster was selected and particles from this class were used for local motion correction following extraction in a 1.48× binned 420×420 pixel box and used for refinement with C2-symmetry imposed giving a final FSC resolution of 2.09 Å resolution for 59,737 particles. This map was further processed by determining the local resolution and during local filtering and sharpening using the local resolution data. For model building, the PhnGHIJ core complex and PhnK from the Phn(GHIJ)$_2$K structure were individually docked into the map using ChimeraX[40] while a PhnL model was generated using Phyre2[31]. The model was further improved by map fitting using Namdinator[33], following correction of the individual chains and addition of ligands and ions using ISOLDE[35]. The model was inspected in Coot[34] where additional residues were added when visible in the map. The final model was refined and validated using Phenix real-space refinement[36].

### Purification of wild type Phn(GHIJKL)$_2$ for cryo-EM

Expression and purification of Phn(GHIJKL)$_2$ with wild type PhnK from pRBS01 followed the same protocol as the PhnK E171Q mutant. After StrepTrap purification, the sample was treated with TEV protease (1:50 w/w TEV protease) at 8 °C overnight, after which the sample was loaded on a MonoQ 5/50 column (Cytiva), pre-equilibrated in 25 mM HEPES/KOH, 100 mM KCl, 5 mM MgCl$_2$, 10% glycerol, 5 mM BME, and 1 µM AMPPNP). The protein was eluted using a gradient of 200–450 mM KCl over 36 CV after which individual peaks were pooled. The Phn(GHIJKL)$_2$ sample was concentrated using Vivaspin 20 filter and loaded on a Superdex 200 increase 10/300 column (Cytiva) equilibrated in 25 mM HEPES/KOH, 125 mM KCl, 0.5 mM EDTA, 5 mM BME, and 0.25 mM ATP. Purified protein was concentrated and stored at 4 °C.

### Cryo-EM sample preparation and analysis of wild type Phn(GHIJKL)$_2$ under ATP turnover conditions

A sample of purified Phn(GHIJKL)$_2$ (2.5 mg/mL) was kept on ice and incubated with ATP at a concentration of 1.5 mM followed by addition of MgCl$_2$ to a concentration of 6 mM to activate the ATPase reaction, then incubated for 15 s at 37 °C and immediately plunge-frozen on glow discharged (10 mA, 90 s, GloQube, Quorum) UltrAuFoil 0.6/1 grids using a Leica plunge freezer EM GP2 set to 37 °C, 100% humidity and an ethane temperature of −184 °C and a blotting time of 6–9 s. Data were acquired in a similar way as for Phn(GHIJKL)$_2$-PhnK E171Q and processed inside cryoSPARC. Micrographs were patch motion corrected, followed by patch CTF estimation, and filtered using various data quality indicators. Particles were initially picked using a gaussian blob picker, after which 4× binned particles were extracted and subjected to 2D classification. Particles from clear 2D classes were kept and a manual picking job using a subset of 14 micrographs was

initiated where more particles were picked. The collected set of picked particles were used to train the Deep picker, after which the model was used to pick particles the remaining 3741 micrographs.

Initial data processing was conducted as for the Phn(GHIJKL)$_2$-Phn E171Q dataset, using 1,155,385 particles from 3741 micrographs. The data processing strategy is shown in Supplementary Figs. 8 and 9. Picked particles were extracted using 2.75× binning and subjected to 2D classification, following selection of good 2D classes. The selected particles from 2D classification were then used for 3D ab initio reconstruction using four classes, and particles from classes showing protein-like features were then used for non-uniform refinement[41]. The resulting particles from the non-uniform refinement were subjected to 3D variability analysis[19] to separate different conformations from each other using three orthogonal principal modes that were determined with resolution filtered to 6 Å, after which particles from different conformations were separated into 8 different clusters by clustering according to the obtained principal modes. For the Phn(GHIJKL)$_2$ WT structure, two clusters were selected, and the corresponding particles were subjected to per-particle local motion correction and reextracted in a 1.29× binned 480 × 480 pixel box. The final set of 222,056 particles was then used for non-uniform refinement using C2 symmetry to gain the final map with a FSC resolution of 1.93 Å (Supplementary Fig. 9a). For the Phn(GHIJK)$_2$ closed conformation, one cluster was selected, and the particles was reextracted in a 1.42× binned 386 × 386 pixel box. The final set of 81,605 particles were used for non-uniform refinement with C2 symmetry, which gave a map with a FSC resolution of 1.98 Å (Supplementary Fig. 9b). For the Phn(GHIJK)$_2$ open conformation, four clusters from the first round of 3D variability analysis were subjected to another round of 3D variability analysis and clustering. One cluster was selected, and the particles were reextracted in a 1.42× binned 386 × 386 pixel box. The final set of 31,280 particles was used for non-uniform refinement and gave a map with a FSC resolution of 2.57 Å (Supplementary Fig. 9c). This map was further processed by determining the local resolution and during local filtering and sharpening of the map using the local resolution data.

For model building of Phn(GHIJKL)$_2$ WT, the final Phn(GHIJKL)$_2$-PhnK E171Q model was initially docked into the map. Ligands were exchanged, waters added using Phenix Douse[36], and the model built manually using Coot and ISOLDE. The final model was refined and validated using Phenix real-space refinement. At the Zn$^{2+}$ binding site in PhnI, ligand bond lengths were restrained using data from C. Lim and T. Dudev, 2000[42]. For the Phn(GHIJK)$_2$ open conformation, the initial model was based on the wildtype Phn(GHIJKL)$_2$ structure, which was split into two parts, one containing PhnG$_2$HI$_2$JK and another containing PhnHJK, each representing one side of the open core complex, and the two docked individually into the map using ChimeraX. The model was rebuilt using ISOLDE and Coot, except for the two PhnK molecules and part of one of the PhnJ molecule, which were not modified due to low local resolution. The final model was refined and validated using Phenix real-space refinement, without the refinement of the PhnK chains. For the Phn(GHIJK)$_2$ closed conformation, the Phn(GHIJKL)$_2$ structure without PhnL was first docked into the map, ligands were exchanged, and waters removed. The final model was refined using Phenix real-space refinement.

### Complementation of Δphn by phnJ, phnK, and phnL mutation

For complementation, a plasmid (pSKA03) containing the entire phn operon including the pho box was constructed based on pBW120 by removal of non-essential sequences. Initially, two large fragments were created using PCR, a plasmid backbone using primers SKA03 and SKA04 and another containing the phn operon including the pho box using primers SKA05 and SKA06. The products were treated with restriction endonuclease DpnI and transformed together (300 ng each) into competent E. coli NovaBlue cells to achieve RecA-independent cloning[43]. Plasmid DNA was purified from positive

clones and confirmed by nucleotide sequencing. Subsequently, the PhnJ E149A, PhnJ Y158A, PhnK R78A/R82A, PhnK E171Q, and PhnL E175Q point mutations were introduced by PCR using primers SKA07 and SKA08 for PhnJ E149A, SKA09 and SKA10 for PhnJ Y158A, SKA11 and SKA12 for PhnK R78A/R82A, SKA13 and SKA14 for PhnK E171Q, and SKA15 and SKA16 for PhnL E175Q. For complementation, the phosphonate knock-out *E. coli* strain HO1488 (Δ*phnHIJKLMNOP*) was used. Strain HO1488 was grown in LB containing 50 µg/mL kanamycin to an $OD_{600}$ of ~0.3, concentrated by centrifugation, and resuspended in 200 µL ice cold TSB (10% w/v PEG 3350, 5% DMSO, and 20 mM $MgCl_2$ in LB media) to make the cells competent for DNA transformation by heat shock with DNA of the plasmids mentioned above. Transformed cells were grown on MOPS minimal agar plates[44] containing 0.2% glucose, 100 µg/mL ampicillin, 34 µg/mL kanamycin, and 0.2 mM of either $K_2HPO_4$, 2-aminoethyl phosphonate, methyl phosphonate or no added phosphate source.

### ATPase assays

Strep-tag purified Phn(GHIJKL)$_2$ complexes (wildtype, PhnK E171Q, PhnL E175Q, and PhnK E171Q + PhnL E175Q) were incubated at a concentration of ~200 nM with 5 mM ATP in 50 mM HEPES/KOH pH 7.5, 300 mM KCl, 10 mM $MgCl_2$, and 5 mM BME at 30 °C overnight. 100 µL of a 1:20 dilution of the reaction mixture was loaded onto a MonoQ column (Cytiva) pre-equilibrated in 25 mM HEPES/KOH pH 7.5, 5 mM $MgCl_2$, and 5 mM BME and eluted using a gradient from 0 to 180 mM KCl over 7 CVs. The column was calibrated with known samples of ATP, ADP, AMP, and adenine to allow identification of potential reaction products. The quantitative rate of ATPase hydrolysis was measured via a coupled enzyme assay conducted at 37 °C in 25 mM HEPES/KOH, 125 mM KCl, 0.5 mM EDTA, 5 mM BME, and 0.25 mM ATP with the addition of 1 mM phosphoenolpyruvate, 350 µM NADH, 0.04 mg/mL pyruvate kinase, 0.1 mg/mL lactate dehydrogenase, 10 mM $MgCl_2$, and ATP to a final concentration of 1 or 5 mM. The reaction was initiated by addition of 0.02 mg StrepTrap-purified wildtype, PhnK E171Q, PhnL E171Q, or PhnK E171Q/PhnL E175Q double mutant in a total volume of 0.35 mL resulting in a final protein concentration of 0.057 mg/mL. Conversion of ATP to ADP was calculated from the measured NADH oxidation rates using an extinction coefficient of 6.2 L mM$^{-1}$ cm$^{-1}$ at 340 nm. The reaction was followed for 6 min, and the reaction rate was measured using a linear fit to the degradation of NADH to NAD$^+$.

### Reporting summary

Further information on research design is available in the Nature Portfolio Reporting Summary linked to this article.

## Data availability

The structural data generated in this study have been deposited in the Protein Data Bank and EM Data Bank under accession codes 7Z19 and EMDB-14445 (Phn(GHIJ)$_2$K), 7Z16 and EMDB-14442 (Phn(GHIJKL)$_2$ PhnK-E171Q:AMPPNP), 7Z15 and EMDB-14441 (Phn(GHIJKL)$_2$ WT:ADP + P$_i$), 7Z18 and EMDB-14444 (Phn(GHIJK)$_2$ WT:ATP closed), and 7Z17 and EMDB-14443 (Phn(GHIJK)$_2$ WT:ATP open). Source data are provided with this paper.

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

## Acknowledgements

This work was made possible through access to the Netherlands Centre for Electron Nanoscopy (NeCEN) at Leiden University, an Instruct-ERIC centre, with assistance from Ludo Renault, the Electron Bio-Imaging Centre (eBIC) at Diamond Light Source, UK, and the Danish National Cryo-EM Facility (EMBION) Aarhus node (iNANO, Aarhus University, Denmark). The authors are also thankful to the Aarhus University Electron Microscopy Computing Cluster (EMCC), Thibaud Louis Antoine Dieudonné for help with the ATPase assay, and Mogens Johannsen and Camilla Bak Nielsen for help with UPLC-QTOF-MS differential analysis of bound phosphorylated metabolites. Financial support was provided by Instruct-ERIC (PID 1514) and a grant from the Novo Nordisk Foundation (grant no. NNF18OC0030646) to D.E.B.

## Author contributions

S.K.A., S.C.O., N.S., B.H.J., and D.E.B. designed and S.K.A, S.C.O., N.S., J.J.E., and R.B.S. carried out the experiments; N.S., S.K.A., and T.B. collected EM data; N.S. determined the initial EM structure of Phn(GHIJ)₂K to lower resolution with help from J.L.K., while S.K.A. determined all high-resolution structures and carried out structure refinement and validation; S.K.A. and D.E.B. wrote the manuscript draft and produced figures, and all authors revised the text.

## Competing interests

The authors declare no competing interests.
