## [Peer Review File · Nature Communications]

Structural remodelling of the carbon-phosphorus lyase machinery by a dual ABC ATPaseREVIEWER COMMENTS

Reviewer #1 (Remarks to the Author):

C-P lyase cleaves the C-P bond in a variety of phosphonates to extract phosphorus, which is an essential nutrient for all microorganisms. The detailed mechanism of how the C-P lyase complex breakdown phosphonates in bacteria cannot be explained by the previous structure of a PhnGHIJ core complex. In this manuscript, the authors presented several high-resolution cryo-EM structures of the C-P lyase core complex PhnGHIJ bound to the ABC ATPase PhnK and PhnL. Surprisingly, the core complex is shown to simultaneously bind a double dimer of PhnK and L in a symmetrical way. Notably, the authors also identified a subpopulation of the complex is present in an asymmetric form with one copy of the PhnJ wide open, which was proposed by the authors to be caused by the ATP hydrolysis that occurred in the ABC ATPase and can potentially open the active site for the substrates. The structures are impressive. However, I have the following concerns that need the authors to address, before I can support its publication in Nature Communications.

My major concern is how physiologically relevant are the symmetrical PhnGHIJKL structures? The authors used either ATPase deficient ABC proteins (PhnK E171Q) with non-hydrolyzable ATP analogues or the WT PhnGHIJKL with high Mg concentrations to initiate ATP hydrolysis. Do these symmetrical PhnGHIJKL complexes also exist for WT PhnGHIJKL under lower Mg concentration as in the E. coli cytosol? Maybe to show in cryo-EM 2D averages that under normal conditions for the WT PhnGHIJKL, such symmetrical complexes also exist. The same argument also holds for the state in which PhnJ is wide open. Will this be an artifact of high Mg concentrations?

I also have the following comments would like the authors to address:

Line 106, “we observe no structural changes in the core complex upon binding of a single subunit PhnK that could explain the functional discrepancies presented by the structure of the C-P lyase core complex.”

This sentence is confusing. What functional discrepancies are the authors referring to?

The authors claim in the text, “no structural changes in the core complex upon binding of a single subunit PhnK”, a figure showing the RMSD of the core complexes before and after PhnK binding is necessary to justify such claim.

Line 136, “compared to the unbound state” is not clear. Do the authors mean, the nucleotide-free state or the state with the core complex bound to only one PhnK as shown in the referenced SFig. 5c?

Line 191, should “PhnK W171Q” be “PhnK E171Q”?

Line 205, Can the author provide more information on why the open state of PhnL should be the ADP state?

SFig. 11b, the model of PRcP into the EM density is not convincing. Since the identity of this molecule is being claimed from the density, it needs to be a better fit. However, in the figure, some atoms of the claimed PRcP are still outside the density.

Reviewer #2 (Remarks to the Author):

In this manuscript Brodersen and colleagues have determined the three-dimensional structures of the Phn complex under a variety of conditions. Significantly, they have shown that the (PhnGHIJ)₂ core complex can bind two copies of PhnK and PhnL. They have also shown that the hydrolysis of ATP by PhnK and PhnL is required for phosphonate metabolism in vivo. Most significantly, the authors have captured a post-ATP hydrolyzed complex that illustrates a dramatic set of conformational changes in the (PhnGHIJ)₂ core complex that have not previously been observed. These are significant advances toward a greater understanding of the very complex catalytic machinery that is utilized to metabolize phosphonates to phosphate products. There are, however, a number of issues that need to be addressed by the authors.

1. In Figure 3 the authors have attempted to measure the ATPase activity of the (PhnGHijkl)₂ complex. In an overnight assay starting with 5 mM ATP they find that approximately 70% of the initial ATP was hydrolyzed. However, with the PhnK mutant only about 20% of the original ATP was hydrolyzed (Figure 2B). Yet the authors conclude on page 10 that “experiments revealed a solid ATPase activity for the wild-type complex, which is almost absent in the PhnK E171Q mutant, suggesting that only PhnK and not PhnL, can hydrolyse ATP under these conditions”. This statement is not fully consistent with the data that is presented in Figure 2B since this figure clearly shows that a significant amount of ATP is hydrolyzed by the PhnK E171Q mutant. Therefore, it is possible to conclude that this level of hydrolysis is due to the ATPase activity of PhnL. To address this issue more clearly, the authors should be asked to conduct the same experiments with the PhnL E175Q mutant as well as the double mutant with PhnK E171Q.
2. In Figure 3C the authors report the catalytic activity of the Phn complex using either 1 or 2 mM ATP. This is a superficial analysis. To more fully understand the ATPase activity of these complexes, the authors should be requested to determine the values of k_{cat} and k_{cat}/K_m for the hydrolysis of ATP. This

should not be too difficult to measure with the coupled kinetic assays using pyruvate kinase and lactate dehydrogenase.

3. If the authors believe that the purification tag on PhnK is somehow preventing the hydrolysis of ATP by PhnL, then the tag should be removed, or they should put the tag somewhere else (page 16). In addition, perhaps the authors could also discuss whether the purification tag is further perturbing the structures that they have reported.

4. In Figure 4B and elsewhere, they show that the “Histidine-site” contains the cyclic phosphate product (PRcP). They need to more clearly explain how this compound is found in the active site since phosphonates were not used during the expression of the genes for the Phn complex. How then is this compound formed? Can they confirm the presence of this compound via mass spectrometry after denaturation of the protein complex?

5. The rearrangement and opening of the core complex is a significant finding, and the authors should be congratulated. However, they have not addressed whether the observed conformational changes result in any modification to the distance between the putative iron-sulfur center and Gly32.

6. In the structures reported here it would be of interest to know the status of the 4 cysteine residues in PhnJ that have been reported to be important for catalysis. Are these residues complexed with a metal ion? Can the binding of metals at this site influence the overall structure of the Phn complex?

7. In the “His site” they have observed the binding of Zn^{2+} . How have the authors determined that the bound metal is actually zinc and not some other divalent cation? Are they certain that the physiological metal is Zn^{2+} and not some other metal ion such as Fe^{2+} ? If the histidine site is suggested at the actual site of P-C bond cleavage it is hard to understand how a radical-based mechanism could be operating with zinc as the focal point. What efforts have been conducted to express the enzyme in the presence of Fe or other cations at this site?

Reviewer #3 (Remarks to the Author):

Please see uploaded review file containing comments to the authors.

While I do some structural biology in my research, please note I am not a structural biologist by training and thus cannot give an in-depth critique of the cryo-EM methodology/data analysis.

The manuscript by Amstrup et al reports a new cryo-EM structure of the core of the C-P lyase complex, showing PhnJ is bound to a unique double dimer comprised of PhnK and PhnL (Phn(GHIJKL)₂). PhnK has bound nucleotide in both active site pockets, whereas PhnL is in an apo form. This is a novel arrangement of two ABC ATPase subunits, and the structure builds on the knowledge of the Phn(GHIJ)₂ structure published by the same group previously and the other cryo-EM structure in this paper, showing flexible attachment of a single PhnK monomer to the enzyme core. The paper nicely describes the interactions between PhnJ and PhnK, and PhnK and PhnL, and further open and closed structures shows ADP and Pi bound by PhnK and nucleotide present in PhnL, as well as a complex lacking PhnL, which appears to be unstable indicating that PhnL has a stabilising role. In one open structure, one of the PhnK subunits moves away from the other, taking PhnJ and H with it and exposing the Zn-binding site at the PhnI-J interface, which has consequences for the coordination of a novel reaction intermediate and further explains why the complex must be a dimer.

The structural work is elegantly complemented with simple *in vivo* growth experiments using an *E. coli* Δphn mutant, confirming the importance of PhnK binding to the core complex and the requirement for ATPase activity of PhnK and PhnL. ATP hydrolysis by the WT complex was also confirmed with *in vitro* ATPase assays; this activity was lost in the PhnK E171Q mutant.

The paper is nicely written, and the data well explained, aided by beautiful figures. The C-P lyase is a fascinating multi-subunit complex with implications from environmental microbiology to biotechnology. This paper will help to understand the catalytic process of this important enzyme, thus I think the paper is highly suitable for publication in Nature Communications and have only relatively minor comments/queries/suggestions that I think would improve the readability and clarity of this very nice manuscript, which are listed below:

Okay to cite papers in the abstract in *Nature Communications*?

Line 9 – sentence structure – better to say: Here we use cryogenic electron microscopy to show..

Line 14 – ABCs is a bit colloquial

Line 18- comma after proteobacterium not needed?

Line 25 – the genes in parenthesis should be changed to proteins? (i.e., *phnCDE* to PhnCDE)?

Line 50 – phosphonates

Line 57 – intro finished a little abruptly – some description of what this paper shows (one line) perhaps?

Line 63 - I know the abbreviated cryo-EM is hyphenated but cryogenic-electron microscopy should not be? (i.e., cryogenic electron microscopy – same on line 9 – see earlier comment)

Line 73 – abbreviation of CID already given above

Lines 75/76 – remove gaps between residues and numbers for Arg78, Tyr158 and Tyr118

Figure 1B - perhaps just my eyes/screen but the white colour does not show up well on a white background

Figure 1C – some of the highlighted residues are not described in the text

Line 91 – Surface should have capital S

Line 97 – CID already given in full earlier in legend (line 93).

Figures 1E and F – what is the significance of the indicated residues in PhnK and Sav1866?

General comment for figures 1 and 2 – the quality of the structures speaks for themselves but might be nice to include SDS-PAGE gels of the complexes in the SI (not essential but if the gels exist).

Figure 2A – Label says ATP but is AMPPNP was added – is ATP modelled into the density of AMPPNP in this case?

Figure 3A – the experimental set up is clear but the plate image is a bit dark (at least on my screen) – colour image would be better?

There is no mention of PhnK-PhnK or PhnL-PhnL inter-dimer interactions – any significant interactions at the dimer interface?

Line 173 – the sentence starting on this line is confusing. It is rather long and although I know what the authors mean it should be re-written to make it clearer for the reader.

Line 177 – solid is not an appropriate adjective for ATPase activity

Figure 3B – the PhnGHIJ₂K result is not described in the text (less active than the E175Q mutant in the complex with KL dimer – evidence need dimer of K or KL?). This should be rectified.

Also, did the authors measure *in vitro* ATPase activity of the complex with PhnL E175Q mutation, which was not functional *in vivo*? Would show if PhnK can hydrolyse ATP without PhnL *in vitro*? I would not make this essential for the paper to be published but the result would be interesting perhaps?

Line 188 – the complex descriptions in the legend do not match those in figure 2B (e.g., use of brackets).

Line 191 – Typo, W171Q should be E171Q. Also subscript 2 missing after PhnGHIJKL.

Line 236 – ATP-bound state?

Line 246 – I think my lack of understanding, but I don't follow the rationale for the presence of the PRcP C-P lyase reaction intermediate if phosphonate was absent? Is this due to a side-reaction with unnatural substrate by the C-P lyase?

Line 275- just to confirm eversion is the right word here?

Line 283 and SI figure 11e – if PhnL dimer closure is prevented by the bulky dual Strep tag on PhnK why was it not cleaved (TEV cleavage site is present)? This should be mentioned earlier when stating that only PhnK has ATPase activity *in vitro* (i.e., it is unlikely that PhnL can bind nucleotide in the purified complex)?

Line 295 – does PhnK here mean the PhnK in the other monomer?

Line 365 – based on comment about the tag and the fact PhnL ATPase activity is needed *in vivo*, can the authors claim only can ABC NBD has activity *in vitro*?

Figure 6 and associated text – I found this model hard to follow, especially in combination with Figure S1. I think the authors have earned the right to speculate but this part of the manuscript was difficult to digest.

Line 571 – 23,400 not 23.400

Line 638 – remove asterisks

Line 690 – delta symbol should not be in italics

Reviewer #4 (Remarks to the Author):

In this contribution by Amstrup et al., the detailed structural analysis of CP lyase machinery is reported.

This is a very interesting story, but unfortunately it raises more questions than gives answers, and in my opinion a significant revision is required to reach conclusive results.

My main concern is that some of the states observed in EM experiments might be artefacts of the complex instability during the sample prep. This relates mostly to states where K and L subunits are very dynamic to the extent that in some states L becomes invisible either due to dissociation or to extreme dynamics. The disagreement observed between in vivo and in vitro experiments makes this concern valid. I believe these structures may serve as a solid base to generate the hypothesis - e.g. the proposed opening of K subunit to expose the catalytic site sounds as a possibility, but the current data do not really prove it. To confirm this, single molecule FRET might be a good option (or EPR) as it can easily confirm whether indeed this opening takes place and that this is not an artefact. If these experimental methods are not accessible at least molecular dynamics simulations should be performed.

The second largest concern is that this manuscript explains very little why the subunit L is necessary. Clearly this is a unique arrangement of proteins never observed before, but the authors failed to clearly explain the need of such a combination and the way how communication between subunits takes place. Their suggestion that subunit L 'serves to control phosphate release' is not supported by any experiments. If this is indeed the case I believe they should see the difference in the phosphate generation assay between the complex with K subunit only and the complex with both K and L present. Also what I miss is K_d values measurements for nucleotide binding for both subunits - are they different? If so, it gives some more information how this system can be regulated. Also I wonder why experiments presented in Fig 3, panels B and C, were not done for E175Q mutant in PhnL. Also how similar are both subunits? Can L substitute K if necessary? How strong are interactions between two? If expressed alone - does phnL has an ATPase activity?

The third largest concern is the role of zinc. Given its proposed importance - I would expect some mutagenesis to be done to see the possible functional and structural impacts.

To wrap up - I find it a very interesting study, yet somewhat preliminary for the publication in a current form.

RESPONSE TO REVIEWER COMMENTS

Reviewer #1 (Remarks to the Author):

C-P lyase cleaves the C-P bond in a variety of phosphonates to extract phosphorus, which is an essential nutrient for all microorganisms. The detailed mechanism of how the C-P lyase complex breakdown phosphonates in bacteria cannot be explained by the previous structure of a PhnGHIJ core complex. In this manuscript, the authors presented several high-resolution cryo-EM structures of the C-P lyase core complex PhnGHIJ bound to the ABC ATPase PhnK and PhnL. Surprisingly, the core complex is shown to simultaneously bind a double dimer of PhnK and L in a symmetrical way. Notably, the authors also identified a subpopulation of the complex is present in an asymmetric form with one copy of the PhnJ wide open, which was proposed by the authors to be caused by the ATP hydrolysis that occurred in the ABC ATPase and can potentially open the active site for the substrates. The structures are impressive. However, I have the following concerns that need the authors to address, before I can support its publication in Nature Communications.

My major concern is how physiologically relevant are the symmetrical PhnGHIJKL structures? The authors used either ATPase deficient ABC proteins (PhnK E171Q) with non-hydrolyzable ATP analogues or the WT PhnGHIJKL with high Mg concentrations to initiate ATP hydrolysis. Do these symmetrical PhnGHIJKL complexes also exist for WT PhnGHIJKL under lower Mg concentration as in the E. coli cytosol? Maybe to show in cryo-EM 2D averages that under normal conditions for the WT PhnGHIJKL, such symmetrical complexes also exist. The same argument also holds for the state in which PhnJ is wide open. Will this be an artifact of high Mg concentrations?

We understand the reviewer's concern for whether the structures obtained are physiologically relevant given the PhnK active site (E171Q) mutant and buffer conditions. Several arguments suggest that they are indeed. First, we don't believe that the choice of Mg²⁺ concentration (6 mM) is unreasonably high. Recall that it must counteract the ion-scavenging effect of the EDTA already present in the sample as well as coordinate ATP (1.5 mM) to make ATP:Mg²⁺. We are therefore not convinced that lowering the Mg²⁺ concentration would represent a more physiologically relevant situation. Finally, with our detailed knowledge of the structures, there is nothing to suggest that they are depending on Mg²⁺ for conformation or stoichiometry, so we find it unlikely that lowering the Mg²⁺ concentration would result in a different set of structures. Finally, we have collected negative stain images of the PhnGHIJKL WT complex in the absence of Mg²⁺. Here, in the presence of ATP, we see clear 2D class averages and 3D volumes that very much resemble the Phn(GHIJKL)₂ structure in the paper.

I also have the following comments would like the authors to address:

Line 106, "we observe no structural changes in the core complex upon binding of a single subunit PhnK that could explain the functional discrepancies presented by the structure of the C-P lyase core complex." This sentence is confusing. What functional discrepancies are the authors referring to?

This comment refers to the discrepancies between the details of the catalytic mechanism as developed through functional studies and our previous crystal structure of the C-P lyase core complex. We have removed the reference to "functional discrepancies" in this context. This aspect is alluded to in the introduction: "An intriguing feature of this structure is that the site of the Fe₄S₄ cluster, presumably required for radical formation and enzyme activation, is located at a significant distance (30 Å) from a glycine residue in PhnJ, which was shown by deuterium exchange studies to be the stable location of a radical between enzymatic turnover cycles.¹"

The authors claim in the text, "no structural changes in the core complex upon binding of a single subunit PhnK", a figure showing the RMSD of the core complexes before and after PhnK binding is necessary to justify such claim.

We thank the reviewer for this, very reasonable, suggestion. We have now included structural overlays of both the Phn(GHIJ)₂ and Phn(GHIJ)₂K structures with rmsd (all atoms) indicated (Supplementary Figure 2c) as well as added rmsd to the existing comparison between Phn(GHIJ)₂K and Phn(GHIJKL)₂ in Supplementary Figure 5e.

Line 136, "compared to the unbound state" is not clear. Do the authors mean, the nucleotide-free state or the state with the core complex bound to only one PhnK as shown in the referenced SFig. 5c?

We agree that this is not clear and have rephrased to "compared to when a single PhnK subunit is bound (rmsd=0.3 Å, Supplementary Fig. 5e)".

Line 191, should "PhnK W171Q" be "PhnK E171Q"?

Thank you! This has been corrected.

Line 205, Can the author provide more information on why the open state of PhnL should be the ADP state?

This is an important point. We only state that the open state mimics an ADP-like state (as these states are open in other ABC systems, i.e. transporters) but not that we think ADP is present in the structure. This has been made clearer in the revision.

SFig. 11b, the model of PRcP into the EM density is not convincing. Since the identity of this molecule is being claimed from the density, it needs to be a better fit. However, in the figure, some atoms of the claimed PRcP are still outside the density.

To address this question, we have updated Supplementary Fig. 11b to show two, perpendicular views of the bound molecule to better visualize how it fits in the density. We additionally attempted to extract small molecules from a protein sample and identify the compound by reversed polarity differential mass spectrometry, however this unfortunately yielded no conclusive results. We have therefore rephrased the text to say: "Based on inspection of the EM density we were able to model the ligand as 5-phospho- α -D-ribose-1,2-cyclic-phosphate (PRcP, Supplementary Fig. 1 and 11b).^{2,3} While this is an intermediate of the C-P lyase pathway, protein expression was carried out in the absence of phosphonates and we therefore believe it could have been formed in a side reaction during overexpression in the absence of phosphonate and was carried along during purification. Due to the minute amounts of the compound present (two molecules per 350 kDa complex), attempts to identify it by UPLC-QTOF-MS differential analysis were not successful."

Reviewer #2 (Remarks to the Author):

In this manuscript Brodersen and colleagues have determined the three-dimensional structures of the Phn complex under a variety of conditions. Significantly, they have shown that the (PhnGHIJ)₂ core complex can bind two copies of PhnK and PhnL. They have also shown that the hydrolysis of ATP by PhnK and PhnL is required for phosphonate metabolism in vivo. Most significantly, the authors have captured a post-ATP hydrolyzed complex that illustrates a dramatic set of conformational changes in the (PhnGHIJ)₂ core complex that have not previously been observed. These are significant advances toward a greater understanding of the very complex catalytic machinery that is utilized to metabolize phosphonates to phosphate products. There are, however, a number of issues that need to be addressed by the authors.

1. In Figure 3 the authors have attempted to measure the ATPase activity of the (PhnGHIJKL)₂ complex. In an overnight assay starting with 5 mM ATP they find that approximately 70% of the initial ATP was hydrolyzed. However, with the PhnK mutant only about 20% of the original ATP was hydrolyzed (Figure 2B). Yet the authors conclude on page 10 that "experiments revealed a solid ATPase activity for the wild-type complex, which is almost absent in the PhnK E171Q mutant, suggesting that only PhnK and not PhnL, can hydrolyse ATP under these conditions". This statement is not fully consistent with the data that is presented in Figure 2B since this figure clearly shows that a significant amount of ATP is hydrolyzed by the PhnK E171Q mutant. Therefore, it is possible to conclude that this level of hydrolysis is due to the ATPase activity of PhnL. To address this issue more clearly, the authors should be asked to conduct the same experiments with the PhnL E175Q mutant as well as the double mutant with PhnK E171Q.

We thank the reviewer for pointing out this discrepancy. In our revised manuscript we include data for the suggested PhnL E175Q mutant, which reveals that both ATP modules need to be active for ATP hydrolysis to take place to the extent observed in wildtype. With either subunit mutated there appears to be a small, residual activity, which is completely gone in the double mutant. This is an important result, and we thank the reviewer for suggesting this experiment. We have adapted the manuscript accordingly throughout.

2. In Figure 3C the authors report the catalytic activity of the Phn complex using either 1 or 2 mM ATP. This is a superficial analysis. To more fully understand the ATPase activity of these complexes, the authors should be requested to determine the values of k_{cat} and k_{cat}/K_m for the hydrolysis of ATP. This should not be too difficult to measure with the coupled kinetic assays using pyruvate kinase and lactate dehydrogenase.

We appreciate that the analysis could be extended by measuring the full Michaelis-Menten kinetics, however, we don't believe that these values are essential for the conclusions of the paper.

3. If the authors believe that the purification tag on PhnK is somehow preventing the hydrolysis of ATP by PhnL, then the tag should be removed, or they should put the tag somewhere else (page 16). In addition, perhaps the authors could also discuss whether the purification tag is further perturbing the structures that they have reported.

As mentioned above, we can now show that both PhnK and PhnL need to be active for ATP hydrolysis to occur, suggesting that the hypothesis involving the purification tag was incorrect. We have therefore removed this suggestion from the revised manuscript and updated the text throughout to reflect this.

4. In Figure 4B and elsewhere, they show that the "Histidine-site" contains the cyclic phosphate product (PRcP). They need to more clearly explain how this compound is found in the active site since phosphonates were not used during the expression of the genes for the Phn complex. How then is this compound formed? Can they confirm the presence of this compound via mass spectrometry after denaturation of the protein complex?

We agree that the presence of this compound is surprising. To address the reviewer's concern, we extracted the compound by denaturation of purified protein complex and analysed this by UPLC-QTOF-MS differential analysis. However, we were unable to identify the compound by this technique, possibly due to the minute amounts present (2 molecules per 350 kDa protein complex). We are thus not able to identify the compound and have adapted the description and conclusions in the revised paper accordingly: " Based on inspection of the EM density we were able to model the ligand as 5-phospho- α -D-ribose-1,2-cyclic-phosphate (PRcP, Supplementary Fig. 1 and 11b).^{2,3} While this is an intermediate of the C-P lyase pathway, protein expression was carried out in the absence of phosphonates and we therefore believe it could have been formed in a side reaction during overexpression in the absence of phosphonate and was carried along during purification. Due to the minute amounts of the compound present (two molecules per 350 kDa complex), attempts to identify it by UPLC-QTOF-MS differential analysis were not successful."

5. The rearrangement and opening of the core complex is a significant finding, and the authors should be congratulated. However, they have not addressed whether the observed conformational changes result in any modification to the distance between the putative iron-sulfur center and Gly32.

This is an important point. We don't see any change in the distance between the iron-sulphur cluster and Gly32 as both elements are in PhnJ, which moves as a rigid body during opening. However, upon opening of the complex, the Gly32 is not far from the putative active site pocket. We thank the reviewer for drawing our attention to this and have included a brief mention of this in the revision as well as one additional figure (Supplementary Fig. 11e) to illustrate this important point.

6. In the structures reported here it would be of interest to know the status of the 4 cysteine residues in PhnJ that have been reported to be important for catalysis. Are these residues complexed with a metal ion? Can the binding of metals at this site influence the overall structure of the Phn complex?

We agree completely with the reviewer that this is important for eventually understanding catalysis and/or activation of C-P lyase. As above, however, we don't see any change in the conformation of the iron-sulphur cluster site between the reported structures and have made this clearer in the revised manuscript.

7. In the "His site" they have observed the binding of Zn²⁺. How have the authors determined that the bound metal is actually zinc and not some other divalent cation? Are they certain that the physiological metal is Zn²⁺ and not some other metal ion such as Fe²⁺? If the histidine site is suggested at the actual site of P-C bond cleavage it is hard to understand how a radical-based mechanism could be operating with zinc as the focal point. What efforts have been conducted to express the enzyme in the presence of Fe or other cations at this site?

In our paper from 2015 (Seweryn et al., Nature, 2015), we demonstrated using anomalous scattering that this ion is likely a Zn²⁺ ion. Moreover, in the present structures, we see a clear transition between a tetrahedral and an octahedral conformation, which to our knowledge, for ions relevant in biology, is only observed for Zn²⁺. Based on this, we feel confident that the ions present in the structures (iron-sulphur site as well as "His site") are indeed zinc. But as for any purified, recombinant protein, we cannot be 100% sure that this also represents the physiological ion, however, the structural transitions as well as presence of a substrate-like molecule suggests this is the case for the His site.

Seweryn, P., Van, L. B., Kjeldgaard, M., Russo, C. J., Passmore, L. A., Hove-Jensen, B., Jochimsen, B., and Brodersen, D. E. (2015) Structural insights into the bacterial carbon-phosphorus lyase machinery. *Nature* 525, 68-72

Reviewer #3 (Remarks to the Author):

While I do some structural biology in my research, please note I am not a structural biologist by training and thus cannot give an in-depth critique of the cryo-EM methodology/data analysis.

The manuscript by Amstrup et al reports a new cryo-EM structure of the core of the C-P lyase complex, showing PhnJ is bound to a unique double dimer comprised of PhnK and PhnL (Phn(GHIJKL)₂). PhnK has bound nucleotide in both active site pockets, whereas PhnL is in an apo form. This is a novel arrangement of two ABC ATPase subunits, and the structure builds on the knowledge of the Phn(GHIJ)₂ structure published by the same group previously and the other cryo-EM structure in this paper, showing flexible attachment of a single PhnK monomer to the enzyme core. The paper nicely describes the interactions between PhnJ and PhnK, and PhnK and PhnL, and further open and closed structures shows ADP and Pi bound by PhnK and nucleotide present in

PhnL, as well as a complex lacking PhnL, which appears to be unstable indicating that PhnL has a stabilising role. In one open structure, one of the PhnK subunits moves away from the other, taking PhnJ and H with it and exposing the Zn-binding site at the PhnI-J interface, which has consequences for the coordination of a novel reaction intermediate and further explains why the complex must be a dimer.

The structural work is elegantly complemented with simple *in vivo* growth experiments using an *E. coli* Δphn mutant, confirming the importance of PhnK binding to the core complex and the requirement for ATPase activity of PhnK and PhnL. ATP hydrolysis by the WT complex was also confirmed with *in vitro* ATPase assays; this activity was lost in the PhnK E171Q mutant.

The paper is nicely written, and the data well explained, aided by beautiful figures. The C-P lyase is a fascinating multi-subunit complex with implications from environmental microbiology to biotechnology. This paper will help to understand the catalytic process of this important enzyme, thus I think the paper is highly suitable for publication in Nature Communications and have only relatively minor comments/queries/suggestions that I think would improve the readability and clarity of this very nice manuscript, which are listed below:

Okay to cite papers in the abstract in *Nature Communications*?

No. We have removed the citations in the abstract.

Line 9 – sentence structure – better to say: Here we use cryogenic electron microscopy to show..

Corrected.

Line 14 – ABCs is a bit colloquial

We agree. Corrected.

Line 18- comma after proteobacterium not needed?

Corrected.

Line 25 – the genes in parenthesis should be changed to proteins? (i.e., *phnCDE* to PhnCDE)?

Corrected.

Line 50 – phosphonates

We prefer "phosphonate" here as a reference to a general class.

Line 57 – intro finished a little abruptly – some description of what this paper shows (one line) perhaps?

We have included a brief description of what is presented at the end of the introduction as suggested.

Line 63 - I know the abbreviated cryo-EM is hyphenated but cryogenic-electron microscopy should not be? (i.e., cryogenic electron microscopy – same on line 9 – see earlier comment)

Corrected.

Line 73 – abbreviation of CID already given above

Thanks. Corrected.

Lines 75/76 – remove gaps between residues and numbers for Arg78, Tyr158 and Tyr118

Corrected.

Figure 1B - perhaps just my eyes/screen but the white colour does not show up well on a white background

Agreed. We have changed the colour of the white density to a darker grey.

Figure 1C – some of the highlighted residues are not described in the text

True. It's a conscious decision to mention the most important residues in the text with more interactions being available in the figure.

Line 91 – Surface should have capital S

Corrected.

Line 97 – CID already given in full earlier in legend (line 93).

We would like to keep the definition in the first figure legend as well as in the text as per the Nature Communications formatting guidelines.

Figures 1E and F – what is the significance of the indicated residues in PhnK and Sav1866?

We have included a description of these residues, which are important for catalysis, in the legend to Figure 1e and 1f in the revised manuscript.

General comment for figures 1 and 2 – the quality of the structures speaks for themselves but might be nice to include SDS-PAGE gels of the complexes in the SI (not essential but if the gels exist).

We agree with the reviewer that this gel can be useful to compare with the corresponding one for the Phn(GHIJKL)₂ in Supplementary Figure 7b. The requested gel has been included in Supplementary Figure 2a.

Figure 2A – Label says ATP but is AMPPNP was added – is ATP modelled into the density of AMPPNP in this case?

We agree that this is somewhat misleading. The caption refers to the "ATP binding site" in reference to the natural function of the site, even though AMPPNP is bound in this case. The text has been corrected to "Details of the PhnK ATP binding site with the AMPPNP (white/orange) shown".

Figure 3A – the experimental set up is clear but the plate image is a bit dark (at least on my screen) – colour image would be better?

There isn't much colour in the pictures, but we have improved the contrast of the present images.

There is no mention of PhnK-PhnK or PhnL-PhnL inter-dimer interactions – any significant interactions at the dimer interface?

The interactions at the PhnK-PhnK interface are relatively standard for an ABC dimer, so we have left out the description due to space limitations. At the PhnL-PhnL interface, due to the open state, there are very few interactions.

Line 173 – the sentence starting on this line is confusing. It is rather long and although I know what the authors mean it should be re-written to make it clearer for the reader.

The sentence was rewritten.

Line 177 – solid is not an appropriate adjective for ATPase activity

Agreed. The sentence was rewritten in the revised manuscript.

Figure 3B – the PhnGHIJ2K result is not described in the text (less active than the E175Q mutant in the complex with KL dimer – evidence need dimer of K or KL?). This should be rectified.

This is an excellent point, i.e. that the experiment with Phn(GHIJ)2K demonstrates that a dimer of ABC modules is required for ATPase activity. The figure has been remade and now includes the PhnL E175Q mutant, but we have included the data for Phn(GHIJ)2K as Supplementary Figure 7c and mentioned this in the text as well.

Also, did the authors measure *in vitro* ATPase activity of the complex with PhnL E175Q mutation, which was not functional *in vivo*? Would show if PhnK can hydrolyse ATP without PhnL *in vitro*? I would not make this essential for the paper to be published but the result would be interesting perhaps?

This is an important point. We have included this experiment in the revised version, see above.

Line 188 – the complex descriptions in the legend do not match those in figure 2B (e.g., use of brackets).

Corrected.

Line 191 – Typo, W171Q should be E171Q. Also subscript 2 missing after PhnGHIJKL. Line 236 – ATP-bound state?

Corrected.

Line 246 – I think my lack of understanding, but I don't follow the rationale for the presence of the PRcP C-P lyase reaction intermediate if phosphonate was absent? Is this due to a side-reaction with unnatural substrate by the C-P lyase?

We believe that the compound represents an off-pathway intermediate picked up from the *E. coli* cytosol. We agree that this point is not entirely clear and have rephrased the description in the revised manuscript (see above).

Line 275- just to confirm eversion is the right word here?

In our understanding, "eversion2 refers to the process by which a transporter membrane protein inverts from an inside-open/outside closed to inside-closed/outside-open state, which we believe is correct in this case.

Line 283 and SI figure 11e – if PhnL dimer closure is prevented by the bulky dual Strep tag on PhnK why was it not cleaved (TEV cleavage site is present)? This should be mentioned earlier when stating that only PhnK has ATPase activity *in vitro* (i.e., it is unlikely that PhnL can bind nucleotide in the purified complex)?

In our revision we demonstrate that the activity of both PhnK and PhnL are required for ATP hydrolysis, therefore this part has been removed and the section rewritten.

Line 295 – does PhnK here mean the PhnK in the other monomer?

This part has been rephrased in the revision.

Line 365 – based on comment about the tag and the fact PhnL ATPase activity is needed *in vivo*, can the authors claim only can ABC NBD has activity *in vitro*?

We agree that this was somewhat contradictory and in fact our updated *in vitro* data for inactivated PhnL clearly show that both subunits need to be active *in vitro* (as well as *in vivo*). The section has been revised to reflect this.

Figure 6 and associated text – I found this model hard to follow, especially in combination with Figure S1. I think the authors have earned the right to speculate but this part of the manuscript was difficult to digest.

We acknowledge this and have updated the model as well as the text to make it clearer and consistent with the new conclusions regarding the function of PhnL.

Line 571 – 23,400 not 23.400

Corrected.

Line 638 – remove asterisks

Corrected.

Line 690 – delta symbol should not be in italics

Corrected.

Reviewer #4 (Remarks to the Author):

In this contribution by Amstrup et al., the detailed structural analysis of CP lyase machinery is reported. This is a very interesting story, but unfortunately it raises more questions than gives answers, and in my opinion a significant revision is required to reach conclusive results.

My main concern is that some of the states observed in EM experiments might be artefacts of the complex instability during the sample prep. This relates mostly to states where K and L subunits are very dynamic to the extent that in some states L becomes invisible either due to dissociation or to extreme dynamics. The disagreement observed between *in vivo* and *in vitro* experiments makes this concern valid. I believe these structures may serve as a solid base to generate the hypothesis - e.g. the proposed opening of K subunit to expose the catalytic site sounds as a possibility, but the current data do not really prove it. To confirm this, single molecule FRET might be a good option (or EPR) as it can easily confirm whether indeed this opening takes place and that this is not an artefact. If these experimental methods are not accessible at least molecular dynamics simulations should be performed.

In our revised manuscript, we show by two different methods that the complex engages in ATP hydrolysis *in vitro*, and that this is dependent on the canonical ATPase active sites of both PhnK and PhnL. This resolves disagreement between the *in vivo* and *in vitro* situations. Moreover, from extensive structural studies of ABC transporters we know how these domains react upon ATP hydrolysis exactly as shown here, namely through forced dissociation of the two ABCs and concomitant structural changes in the transmembrane part / core complex. We appreciate that FRET could shed additional light on this, but we think that would be beyond the scope of this paper, which focusses on structural studies. Moreover, we believe that MD is not likely to correctly model large-scale structural changes as those shown here.

The second largest concern is that this manuscript explains very little why the subunit L is necessary. Clearly this is a unique arrangement of proteins never observed before, but the authors failed to clearly explain the need of such a combination and the way how communication between subunits takes place. Their suggestion that subunit L 'serves to control phosphate release' is not supported by any experiments. If this is indeed the case I believe they should see the difference in the phosphate generation assay between the complex with K subunit only and the complex with both K and L present.

We appreciate the reviewer's wish to delineate the roles of PhnK and PhnL. In our revised manuscript we demonstrate that the ATPase activity of both PhnK and PhnL are required for a C-P lyase to be functional *in vitro* as well as *in vivo*. This also means that the experiment suggested cannot be carried out as there would be no activity. To accommodate the reviewer's concern, however, we have updated the text and model to reflect what we believe is the reason that there are two ABC dimers in this case. That of course will need verification, but we believe that lies beyond the scope of the present work. Regarding the mentioning of a complex with the PhnK subunit only our observation is that we can only get Phn(GHIJ)2K (i.e. a single PhnK copy) when PhnL is absent. In this complex we do not see any ATPase activity, likely due to the lack of an ABC (PhnK) dimer. The Phn(GHIJK)2 complex that we assume the reviewer refers to here (Supplementary Figure 9b of the revised manuscript), was only observed in the ATP turnover cryo-EM dataset and likely resembles a transient state of the cryo-EM data.

Also what I miss is K_d values measurements for nucleotide binding for both subunits - are they different? If so, it gives some more information how this system can be regulated.

We appreciate the wish for K_d values, but they are non-trivial to determine as PhnK only binds ATP in the presence of PhnL and vice versa. We also do not believe these values are essential for the conclusions in the paper.

Also I wonder why experiments presented in Fig 3, panels B and C, were not done for E175Q mutant in PhnL.

We have included this experiment in the revised version, see above.

Also how similar are both subunits? Can L substitute K if necessary? How strong are interactions between two?

We thank the reviewer for this insightful suggestion. As described in the manuscript, there are distinct differences between PhnK and PhnL that likely define their roles and binding abilities. We have attempted to substitute PhnK for PhnL genetically but observed no binding of the subunit in this case, so our hypothesis is that PhnK evolved to specifically bind the core complex while PhnL evolved to bind PhnK.

If expressed alone - does phnL has an ATPase activity?

In our hands, PhnL is insoluble when expressed in isolation. Previously, Kamat et al reported isolated purification of this subunit using a large GST tag, but no ATPase activity was observed. Upon removal of the GST tag, the protein became insoluble.

Kamat, S. S., Williams, H. J., and Raushel, F. M. (2011) Intermediates in the transformation of phosphonates to phosphate by bacteria. *Nature* 480, 570-573

The third largest concern is the role of zinc. Given its proposed importance - I would expect some mutagenesis to be done to see the possible functional and structural impacts.

Already in our 2015 paper (citation below), we demonstrated using anomalous scattering that this site indeed contains Zn²⁺ and showed by mutagenesis that the histidine residues coordinating the ion are required for phosphonate breakdown *in vivo*. So, we believe that the reviewer's point, which is valid, has been addressed previously.

Seweryn, P., Van, L. B., Kjeldgaard, M., Russo, C. J., Passmore, L. A., Hove-Jensen, B., Jochimsen, B., and Brodersen, D. E. (2015) Structural insights into the bacterial carbon-phosphorus lyase machinery. *Nature* 525, 68-72

To wrap up - I find it a very interesting study, yet somewhat preliminary for the publication in a current form.

REVIEWERS' COMMENTS

Reviewer #1 (Remarks to the Author):

The authors have addressed my minor concerns. However, they still left my major concern unaddressed. The authors mentioned they have performed negative-stain EM of the WT-GHIJKL complex, however, the data are not shown. Nevertheless, negative-stain results cannot be useful since the staining reagent leads to the extremely high-salt condition. What I suggested is just a limited number of cryo-EM images of WT-GHIJKL, which allow for generating 2D class averages to show the symmetrical GHIJKL complex.

Reviewer #2 (Remarks to the Author):

In general, I am satisfied with the modifications made by Amstrup et al. However, I think that the authors need to clarify the three-dimensional coordination geometry of the Zn²⁺ site in the protein complex. The authors state in their revision letter that, “we see a clear transition between a tetrahedral and an octahedral conformation, which to our knowledge, for ions relevant in biology, is only observed for Zn²⁺. The authors should provide further support for this statement by providing a referenced example in enzymology where a site for the binding of Zn²⁺ goes from a tetrahedral geometry to one with an octahedral geometry.

Reviewer #3 (Remarks to the Author):

Having looked through the revised manuscript I am satisfied that the authors have addressed all my queries and congratulate them on a very nice paper.

I spotted one new typo.

Line 70 - this is now the first use of cryogenic electron microscopy (cryo-EM) in the text and so should be defined here in full (without the hyphen between cryogenic and electron!). On line 79 the abbreviated form can now be used i.e., cryo-EM

Reviewer #4 (Remarks to the Author):

I congratulate the authors with the much improved manuscript - in the current form it is more streamlined and the story is more complete - although as I said earlier there are more questions than answers, but now the authors give a clear outlook.

I was somewhat surprised with the notion that in the coupled ATP assay the negative value was obtained for the double mutant, just a glitch? But clearly the double mutant is inactive, so no concerns.

AUTHOR'S RESPONSE TO REVIEWERS' COMMENTS

Reviewer #1 (Remarks to the Author):

The authors have addressed my minor concerns. However, they still left my major concern unaddressed. The authors mentioned they have performed negative-stain EM of the WT-GHIJKL complex, however, the data are not shown. Nevertheless, negative-stain results cannot be useful since the staining reagent leads to the extremely high-salt condition. What I suggested is just a limited number of cryo-EM images of WT-GHIJKL, which allow for generating 2D class averages to show the symmetrical GHIJKL complex.

We believe that the structure of PhnGHIJKL WT in the presence of ADP+Pi (Supplementary Figure 5b) demonstrates that the symmetrical PhnGHIJKL complex is present in the wild type. Moreover, the Mg²⁺ concentration used (6 mM) is not unnaturally high. We hope this addresses the reviewer's concern.

Reviewer #2 (Remarks to the Author):

In general, I am satisfied with the modifications made by Amstrup et al. However, I think that the authors need to clarify the three-dimensional coordination geometry of the Zn²⁺ site in the protein complex. The authors state in their revision letter that, "we see a clear transition between a tetrahedral and an octahedral conformation, which to our knowledge, for ions relevant in biology, is only observed for Zn²⁺. The authors should provide further support for this statement by providing a referenced example in enzymology where a site for the binding of Zn²⁺ goes from a tetrahedral geometry to one with an octahedral geometry.

The support for the transition between the octahedral to tetrahedral conformations of the Zn²⁺ ion comes mainly from C. Lim and T. Dudev, 2000, who find that the octahedral conformation is preferred when fewer protein residues interact with the ion. In our structure, we observe something similar, namely that the octahedral geometry is found when fewer protein residues (and consequently more water molecules) are bound to Zn²⁺ (See Sup. Fig. 11a, octahedral, three protein residues versus Sup. Fig. 11d, tetrahedral, four protein residues). The article by C. Lim and T. Dudev also measures ion-protein interaction distances and was also used as a reference point for setting up restraints for the two structures and is only mentioned in the Methods section.

On the other hand, we were not able to find previous examples where Zn²⁺ changes coordination geometry from tetrahedral to octahedral inside an enzyme, which indeed may be novel and a potentially highly important functional feature of C-P lyase.

Reviewer #3 (Remarks to the Author):

Having looked through the revised manuscript I am satisfied that the authors have addressed all my queries and congratulate them on a very nice paper.

Thank you very much. :-)

I spotted one new typo.

Line 70 - this is now the first use of cryogenic electron microscopy (cryo-EM) in the text and so should be defined here in full (without the hyphen between cryogenic and electron!). On line 79 the abbreviated form can now be used i.e., cryo-EM

Corrected.

Reviewer #4 (Remarks to the Author):

I congratulate the authors with the much improved manuscript - in the current form it is more streamlined and the story is more complete - although as I said earlier there are more questions than answers, but now the authors give a clear outlook.

Thank you!

I was somewhat surprised with the notion that in the coupled ATP assay the negative value was obtained for the double mutant, just a glitch? But clearly the double mutant is inactive, so no concerns.

This data point was removed due to high background.